# Functional reconstitution of a bacterial CO₂ concentrating mechanism in *Escherichia coli*

Avi I Flamholz[1], Eli Dugan[1], Cecilia Blikstad[1], Shmuel Gleizer[2], Roee Ben-Nissan[2], Shira Amram[2], Niv Antonovsky[2†], Sumedha Ravishankar[1‡], Elad Noor[2§], Arren Bar-Even[3], Ron Milo[2]*, David F Savage[1]*

[1]Department of Molecular and Cell Biology, University of California, Berkeley, Berkeley, United States; [2]Department of Plant and Environmental Sciences, Weizmann Institute of Science, Rehovot, Israel; [3]Max Planck Institute of Molecular Plant Physiology, Potsdam, Germany

*For correspondence:
ron.milo@weizmann.ac.il (RM);
savage@berkeley.edu (DFS)

Present address: †Laboratory of Genetically Encoded Small Molecules, The Rockefeller University, New York, United States; ‡Division of Biological Sciences, Section of Molecular Biology, University of California, San Diego, La Jolla, United States; §Institute of Molecular Systems Biology, Eidgenössische Technische Hochschule Zürich, Zürich, Switzerland

**Abstract** Many photosynthetic organisms employ a CO₂ concentrating mechanism (CCM) to increase the rate of CO₂ fixation via the Calvin cycle. CCMs catalyze ≈ 50% of global photosynthesis, yet it remains unclear which genes and proteins are required to produce this complex adaptation. We describe the construction of a functional CCM in a non-native host, achieved by expressing genes from an autotrophic bacterium in an *Escherichia coli* strain engineered to depend on rubisco carboxylation for growth. Expression of 20 CCM genes enabled *E. coli* to grow by fixing CO₂ from ambient air into biomass, with growth in ambient air depending on the components of the CCM. Bacterial CCMs are therefore genetically compact and readily transplanted, rationalizing their presence in diverse bacteria. Reconstitution enabled genetic experiments refining our understanding of the CCM, thereby laying the groundwork for deeper study and engineering of the cell biology supporting CO₂ assimilation in diverse organisms.

## Introduction

Nearly all carbon in the biosphere enters by CO₂ fixation in the Calvin-Benson-Bassham cycle (*Bassham, 2003*; *Bassham et al., 1954*; *Benson, 2002*; *Field et al., 1998*; *Raven, 2009*). Ribulose Bisphosphate Carboxylase/Oxygenase - commonly known as rubisco - is the CO₂ fixing enzyme in this cycle (*Kawashima and Wildman, 1971*; *Weissbach et al., 1956*; *Wildman, 2002*) and likely the most abundant enzyme on Earth (*Bar-On and Milo, 2019*).

As rubisco is abundant and central to biology, one might expect it to be an exceptional catalyst, but it is not. Photosynthetic rubiscos are modest enzymes, with carboxylation turnover numbers ($k_{cat}$) ranging from 1 to 10 s$^{-1}$ (*Badger et al., 1998*; *Flamholz et al., 2019*; *Iñiguez et al., 2020*; *Jordan and Ogren, 1983*; *Savir et al., 2010*; *Tcherkez et al., 2006*). Moreover, all known rubiscos catalyze a competing oxygenation of the five-carbon organic substrate, ribulose 1, 5-bisphosphate (*Bathellier et al., 2018*; *Bowes and Ogren, 1972*; *Cleland et al., 1998*). Rubisco oxygenation represents a 'waste' of cellular resources on two fronts: it fails to generate any new organic carbon and also produces a molecule (2-phosphoglycolate) that is not part of the Calvin cycle and therefore must be recycled through a salvage pathway to keep the cycle going (*Busch, 2020*).

Rubisco arose >2.5 billion years ago, when Earth's atmosphere contained little O₂ and abundant CO₂ (*Fischer et al., 2016*; *Shih et al., 2016*). In this environment, rubisco's eponymous oxygenase activity could not have hindered carbon fixation or the growth of CO₂-fixing organisms. Present-day atmosphere, however, poses a problem for plants and other autotrophs: their primary carbon

source, $CO_2$, is relatively scarce ($\approx 0.04\%$) while a potent competing substrate, $O_2$, is abundant ($\approx 21\%$).

$CO_2$ concentrating mechanisms (CCMs) arose multiple times over the last 2 billion years (*Flamholz and Shih, 2020*; *Raven et al., 2017*) and overcame rubisco's limitations by concentrating $CO_2$ near the enzyme (*Figure 1A*). In an elevated $CO_2$ environment, most rubisco active sites will be occupied with $CO_2$ and not $O_2$. As such, high $CO_2$ is expected to increase the rate of carboxylation and competitively inhibit oxygenation (*Bowes and Ogren, 1972*) thereby improving overall carbon assimilation (*Figure 1B*). Today, at least four varieties of CCMs are found in plants, algae, and bacteria (*Flamholz and Shih, 2020*; *Raven et al., 2017*), organisms with CCMs are collectively responsible for $\approx 50\%$ of global net photosynthesis (*Raven et al., 2017*), and some of the most productive human crops (e.g. maize and sugarcane) rely on CCMs.

CCMs are particularly common among autotrophic bacteria: all Cyanobacteria and many Proteobacteria have CCM genes (*Kerfeld and Melnicki, 2016*; *Rae et al., 2013*). Bacterial CCMs rely on two crucial features: (i) energy-coupled inorganic carbon uptake at the cell membrane and (ii) a 200+ MDa protein organelle called the carboxysome that encapsulates rubisco with a carbonic anhydrase enzyme (*Desmarais et al., 2019*; *Kaplan et al., 1980*; *Price and Badger, 1989a*, *Price and Badger, 1989b*; *Rae et al., 2013*; *Shively et al., 1973*). In the prevailing model of the carboxysome CCM (*Fridlyand et al., 1996*; *Mangan et al., 2016*; *McGrath and Long, 2014*), inorganic carbon uptake produces a high, above-equilibrium cytosolic $HCO_3^-$ concentration ($\approx 30$ mM) that diffuses into the carboxysome, where carbonic anhydrase activity produces a high carboxysomal $CO_2$ concentration that promotes efficient carboxylation by rubisco (*Figure 1A–B*).

As CCMs accelerate $CO_2$ fixation by rubisco, there is great interest in transplanting them into crops (*Ermakova et al., 2020*; *McGrath and Long, 2014*). Carboxysome-based CCMs are especially attractive because they natively function in single cells and appear to rely on a tractable number of genes (*Lin et al., 2014*; *Long et al., 2018*; *Occhialini et al., 2016*; *Orr et al., 2020*). Modeling suggests that introducing bacterial CCM components could improve plant photosynthesis (*McGrath and Long, 2014*), especially if aspects of plant physiology can be modulated via genetic engineering (*Wu et al., 2019*). However, expressing bacterial rubiscos and carboxysome components has, so far, uniformly resulted in transgenic plants displaying impaired growth (*Lin et al., 2014*; *Long et al., 2018*; *Occhialini et al., 2016*; *Orr et al., 2020*). More generally, as our understanding of the genes and proteins participating in the carboxysome CCM rests mostly on loss-of-function genetic experiments in native hosts (*Baker et al., 1998*; *Cai et al., 2009*; *Cannon et al., 2001*; *Desmarais et al., 2019*; *Marcus et al., 1986*; *Ogawa et al., 1987*; *Price and Badger, 1989a*), it is possible that some genetic, biochemical, and physiological aspects of CCM function remain unappreciated. We therefore sought to test whether current understanding is sufficient to reconstitute the bacterial CCM in a non-native bacterial host, namely *Escherichia coli*.

Using a genome-wide screen in the $CO_2$-fixing proteobacterium *Halothiobacillus neapolitanus*, we recently demonstrated that a 20-gene cluster encodes all activities required for the CCM, at least in principle (*Desmarais et al., 2019*). These genes are detailed in *Supplementary file 1* and include rubisco large and small subunits, the carboxysomal carbonic anhydrase, seven structural proteins of the α-carboxysome (*Bonacci et al., 2012*), an energy-coupled inorganic carbon transporter (*Desmarais et al., 2019*; *USF MCB4404L et al., 2017*; *Scott et al., 2019*), three rubisco chaperones (*Aigner et al., 2017*; *Feiz et al., 2014*; *Mueller-Cajar, 2017*; *Wheatley et al., 2014*), and four genes of uncertain function (*Figure 1C*). We aimed to test whether these genes are sufficient to establish a functioning CCM in *E. coli*.

## Results

As *E. coli* is a heterotroph, consuming organic carbon molecules to produce energy and biomass, it does not natively rely on rubisco. Therefore, in order to evaluate the effect of heterologous CCM expression, we first designed an *E. coli* strain that depends on rubisco carboxylation for growth. To grow on glycerol as the sole carbon source, *E. coli* must synthesize ribose 5-phosphate (Ri5P) for nucleic acids. Synthesis of Ri5P via the pentose phosphate pathway forces co-production of ribulose 5-phosphate (Ru5P). Deletion of ribose 5-phosphate isomerase (*rpiAB* genes, denoted Δrpi), however, makes Ru5P a metabolic 'dead-end' (*Figure 2A*). Expression of phosphoribulokinase (*prk*) and rubisco enables a 'detour' pathway converting Ru5P and $CO_2$ into two units of the central

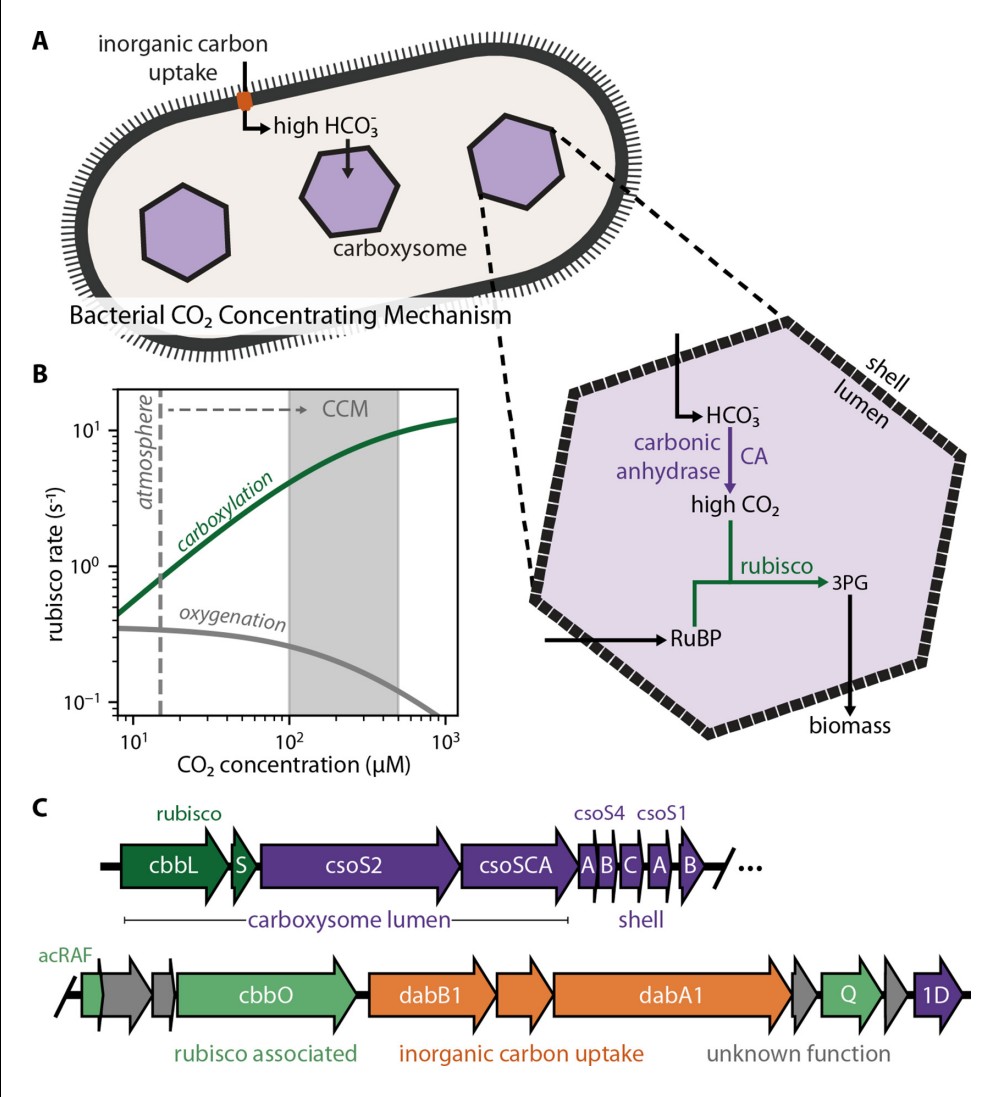

**Figure 1.** Twenty genes form the basis of a bacterial CCM. (**A**) The bacterial CCM consists of at least two essential components - energy-coupled inorganic carbon uptake and carboxysome structures that encapsulate rubisco with a carbonic anhydrase (CA) enzyme (*Desmarais et al., 2019*; *Kaplan et al., 1980*; *Price and Badger, 1989a*, *Price and Badger, 1989b*; *Rae et al., 2013*; *Shively et al., 1973*). Transport generates a large cytosolic $HCO_3^-$ pool, which is rapidly converted to high carboxysomal $CO_2$ concentration by the carboxysomal CA (*Mangan et al., 2016*; *McGrath and Long, 2014*). (**B**) Elevated $CO_2$ increases the rubisco carboxylation rate (green) and suppresses oxygenation by competitive inhibition (grey). [$O_2$] was set to 270 µM for rate calculations. A more detailed version of this calculation is described in *Figure 1—figure supplement 1*. (**C**) *H. neapolitanus* CCM genes are mostly contained in a 20 gene cluster (*Desmarais et al., 2019*) expressing rubisco and its associated chaperones (green), carboxysome structural proteins (purple), and an inorganic carbon transporter (orange). *Supplementary file 1* gives fuller description of the functions of these 20 genes along with a per-gene bibliography. *Figure 1—figure supplement 2* demonstrates that the operon beginning with acRAF indeed encodes a functional inorganic carbon transporter.

The online version of this article includes the following figure supplement(s) for figure 1:

**Figure supplement 1.** Elevated $CO_2$ overcomes limitations associated with rubisco catalysis.

**Figure supplement 2.** The 20 gene CCM cluster includes a functional DAB-type inorganic carbon transporter.

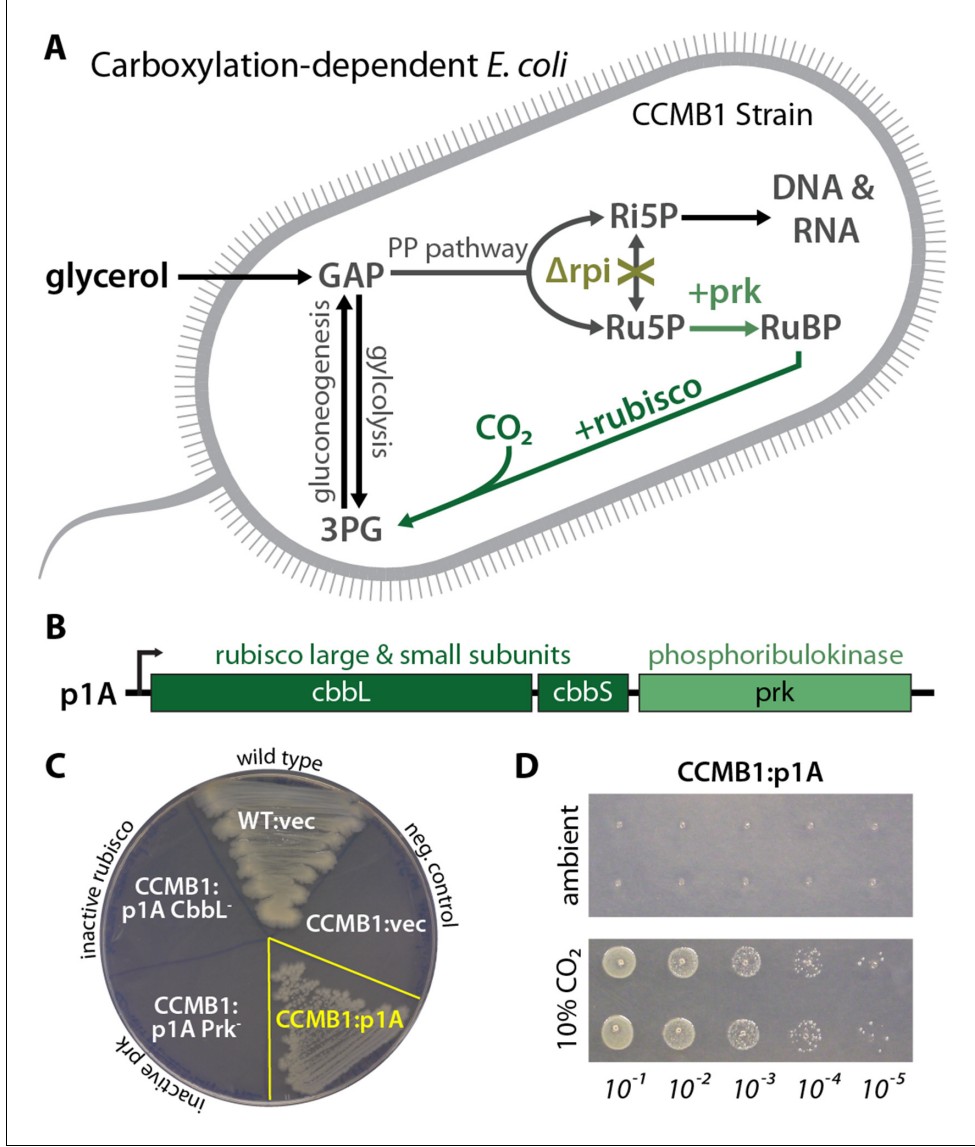

**Figure 2.** CCMB1 depends on rubisco carboxylation for growth on glycerol. (**A**) Ribose-5-phosphate (Ri5P) is required for nucleotide biosynthesis. Deletion of ribose-phosphate isomerase (Δrpi) in CCMB1 blocks ribulose-5-phosphate (Ru5P) metabolism in the pentose phosphate (PP) pathway. Expression of rubisco (*H. neapolitanus* CbbLS) and phosphoribulokinase (*S. elongatus* PCC7942 *prk*) on the p1A plasmid (**B**) permits Ru5P metabolism, thus enabling growth on M9 glycerol media in 10% $CO_2$ (**C**). Mutating the rubisco active site (p1A CbbL⁻) abrogates growth, as does mutating ATP-binding residues of Prk (p1A Prk⁻). (**D**) CCMB1:p1A grows well under 10% $CO_2$, but fails to grow in ambient air. Cells were grown on M9 glycerol media throughout. The algorithmic design of CCMB1 is described in *Figure 2—figure supplement 4* and Appendix 1. The mechanism of rubisco-dependence is diagrammed in *Figure 2—figure supplement 3*. Figure supplement 2 demonstrates growth of CCMB1:p1A on various media, *Figure 2—figure supplement 5* demonstrates complementation by a variety of bacterial rubiscos and *Figure 2—figure supplement 1* demonstrates anaerobic growth of CCMB1:p1A, establishing that oxygenation is not required for growth. Acronyms: ribulose 1, 5-bisphosphate (RuBP), 3-phosphoglycerate (3PG).

The online version of this article includes the following figure supplement(s) for figure 2:

**Figure supplement 1.** Expression of five kinetically and phylogenetically distinct rubiscos permits CCMB1 growth in glycerol minimal media with 5% $CO_2$.

**Figure supplement 2.** CCMB1 does not require oxygen for growth in minimal media.

**Figure supplement 3.** Proposed mechanisms of rubisco-dependent growth in CCMB1.

**Figure supplement 4.** The OptSlope algorithm for designing rubisco-coupled *E.coli*strains.

*Figure 2 continued on next page*

*Figure 2 continued*

**Figure supplement 5.** CCMB1 depends on rubisco and *prk* for growth in minimal media.

metabolite 3-phosphoglycerate (3PG), enabling Ru5P metabolism and growth (*Figure 2A*). Additionally, cytosolic carbonic anhydrase activity is incompatible with the bacterial CCM (*Price and Badger, 1989b*). We therefore constructed a strain, named CCMB1 for 'CCM **B**ackground **1**', lacking *rpiAB* and all endogenous carbonic anhydrases (Materials and methods, Appendix 1).

As predicted, CCMB1 required rubisco and *prk* for growth on glycerol minimal media in 10% $CO_2$ (*Figure 2B–C*). CCMB1:p1A failed to grow on glycerol media in ambient air, however, presumably due to insufficient carboxylation at low $CO_2$ (*Figure 2D*). As such, CCMB1:p1A displays the 'high-$CO_2$ requiring' phenotype that is the hallmark of CCM mutants (*Baker et al., 1998*; *Marcus et al., 1986*; *Price and Badger, 1989a*). Four additional bacterial rubiscos were tested and displayed the same pattern, enabling CCMB1 to grow reproducibly in high $CO_2$ but not in ambient air (*Figure 2—figure supplement 1*). When expressing rubisco and *prk* from the p1A plasmid, CCMB1 also grew reproducibly in an anoxic mix of 10:90 $CO_2$:$N_2$ (*Figure 2—figure supplement 2*) implying that carboxylation is sufficient for growth on glycerol media and rubisco-catalyzed oxygenation of RuBP is not required.

We expected that expressing a functional $CO_2$-concentrating mechanism would cure CCMB1 of its high-$CO_2$ requirement and permit growth in ambient air. We therefore generated two plasmids, pCB and pCCM, that together express all 20 genes from the *H. neapolitanus* CCM cluster (*Figure 1C*). pCB encodes 10 carboxysome genes (*Bonacci et al., 2012*; *Cai et al., 2008*), including rubisco large and small subunits, along with *prk*. The remaining *H. neapolitanus* genes, including putative rubisco chaperones (*Aigner et al., 2017*; *Mueller-Cajar, 2017*; *Wheatley et al., 2014*) and an inorganic carbon transporter (*Desmarais et al., 2019*; *Scott et al., 2019*), were cloned into the second plasmid, pCCM.

CCMB1 co-transformed with pCB and pCCM initially failed to grow on glycerol media. We therefore conducted selection experiments, described fully in Appendix 2, that ultimately resulted in the isolation of mutant plasmids conferring growth in ambient air. Briefly, CCMB1:pCB + pCCM cultures were grown to saturation in 10% $CO_2$. These cultures were washed and plated on glycerol minimal media (Materials and methods). Colonies became visible after 20 days of incubation in ambient air, but only when induction and both plasmids were provided (*Figure 3—figure supplement 1*). Deep-sequencing of plasmid DNA revealed mutations in regulatory sequences (e.g. a promoter and transcriptional repressor) but none in sequences coding for CCM components (*Supplementary file 1*). Individual post-selection plasmids pCB' and pCCM' were reconstructed by PCR, resequenced, and transformed into naive CCMB1 (Materials and methods). As shown in *Figure 3*, pCB' and pCCM' together enabled reproducible growth of CCMB1 in ambient air, suggesting that the 20 genes expressed are sufficient to produce a heterologous CCM without any genomic mutations.

To verify that growth in ambient air depends on the CCM, we generated plasmids carrying targeted mutations to known CCM components (*Figure 4*). An inactivating mutation to the carboxysomal rubisco (CbbL K194M) prohibited growth entirely. Mutations targeting the CCM, rather than rubisco itself, should ablate growth in ambient air while permitting growth in high $CO_2$ (*Desmarais et al., 2019*; *Mangan et al., 2016*; *Marcus et al., 1986*; *Price and Badger, 1989a*; *Rae et al., 2013*). Consistent with this understanding, an inactive mutant of the carboxysomal carbonic anhydrase (CsoSCA C173S) required high-$CO_2$ for growth. Similarly, disruption of carboxysome formation by removal of the pentameric shell proteins or the N-terminal domain of CsoS2 also eliminated growth in ambient air. Removing the pentameric proteins CsoS4AB disrupts the permeability barrier at the carboxysome shell (*Cai et al., 2009*), while truncating CsoS2 prohibits carboxysome formation entirely (*Oltrogge et al., 2020*). Finally, an inactivating mutation to the inorganic carbon transporter also eliminated growth in ambient air (*Desmarais et al., 2019*).

These experiments demonstrate that pCB' and pCCM' enable CCMB1 to grow in ambient air in a manner that depends on the known components of the bacterial CCM. To confirm that these cells produce carboxysome structures, we performed thin section electron microscopy. Regular polyhedral inclusions of $\approx$100 nm diameter were visible in micrographs (*Figure 5A*), implying production of morphologically normal carboxysomes. Furthermore, we were able to purify carboxysome structures

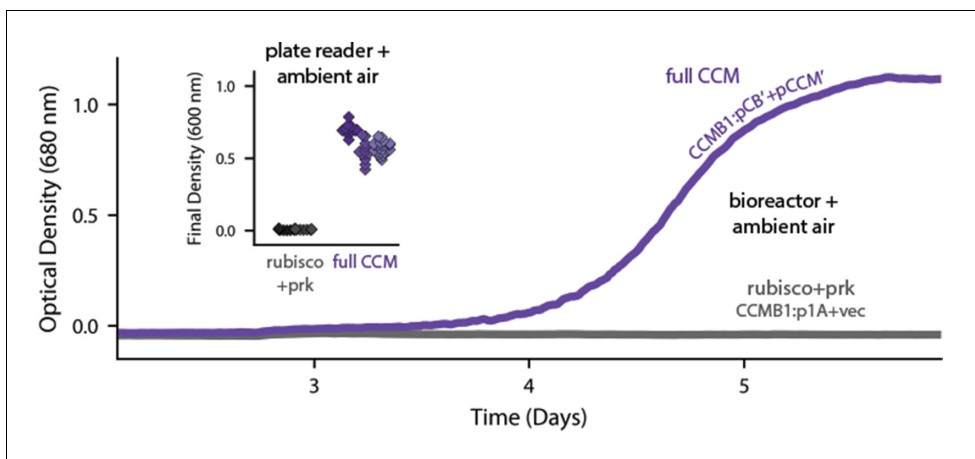

**Figure 3.** Expression of 20 CCM genes permits growth of CCMB1 in ambient air. Time course data give representative growth curves from a bioreactor bubbling ambient air. CCMB1:pCB' + pCCM' grows well (purple, 'full CCM'), while rubisco and *prk* alone are insufficient for growth in ambient air (grey, CCMB1:p1A+vec). Inset: a plate reader experiment in biological triplicate (different shades) gave the same result. Expressing the full complement of CCM genes led to an increase in culture density (optical density at 600 nm) of ≈0.6 units after 80 hr of cultivation. Bootstrapping was used to calculate a 99.9% confidence interval of 0.56–0.64 OD units for the effect of expressing the full CCM during growth in ambient air. *Figure 3—figure supplement 1* and Appendix 2 describe the selection procedures in detail while *Figure 3—figure supplement 2* shows triplicate growth curves and evaluates statistical significance.

The online version of this article includes the following figure supplement(s) for figure 3:

**Figure supplement 1.** A series of selection experiments produced mutant plasmids that permit rubisco-dependent growth in ambient air.

**Figure supplement 2.** pCB' and pCCM' permit CCMB1 to grow in ambient air.

---

from CCMB1:pCB'+pCCM' using established methods. Carboxysomes from CCMB1:pCB'+pCCM' were similar in appearance to those from the native host, although more heterogeneous in size and shape (*Figure 5B*). The rubisco complex was visible inside isolated carboxysomes and confirmed to co-migrate with the structure via SDS-PAGE analysis (*Figure 5—figure supplement 1*).

We next conducted isotopic labeling experiments to determine whether CCMB1:pCB' + pCCM' fixes $CO_2$ from ambient air into biomass. Cells were grown in minimal media with $^{13}C$-labeled glycerol as the sole organic carbon source, such that $CO_2$ from ambient air was the dominant source of $^{12}C$. The isotopic composition of amino acids in total biomass hydrolysate was analyzed via mass spectrometry (Materials and methods). Serine is a useful sentinel of rubisco activity because *E. coli* produces it from the rubisco product 3PG (*Stauffer, 2004*; *Szyperski, 1995*). 3PG is also an intermediate of lower glycolysis (*Bar-Even et al., 2012*), and so the degree of $^{12}C$ labeling on serine reports on the balance of fluxes through rubisco and lower glycolysis (*Figure 6A*). We therefore expected excess $^{12}C$ labeling of serine when rubisco is active in CCMB1. Consistent with this expectation, serine from CCMB1:pCB'+pCCM' cells contained roughly threefold more $^{12}C$ than the rubisco-independent control (*Figure 6B*). We estimated the contribution of rubisco to 3PG synthesis *in vivo* by comparing labeling patterns between the rubisco-dependent experimental cultures and controls (Appendix 2). Based on these estimates, rubisco carboxylation was responsible for at least 10% of 3PG synthesis in all four biological replicates (*Figure 6C*, Materials and methods), confirming fixation of $CO_2$ from ambient air. As such, this work represents the first functional reconstitution of any CCM.

Reconstitution in *E. coli* enabled us to investigate which *H. neapolitanus* genes are necessary for CCM function in the absence of any regulation or genetic redundancy (i.e. genes with overlapping function) present in the native host. We focused on genes involved in rubisco proteostasis and generated plasmids lacking *acRAF*, a putative rubisco chaperone, or carrying targeted mutations to CbbQ, an ATPase involved in activating rubisco catalysis (*Aigner et al., 2017*; *Mueller-Cajar, 2017*; *Sutter et al., 2015*; *Wheatley et al., 2014*). Although *acRAF* deletion had a large negative effect in

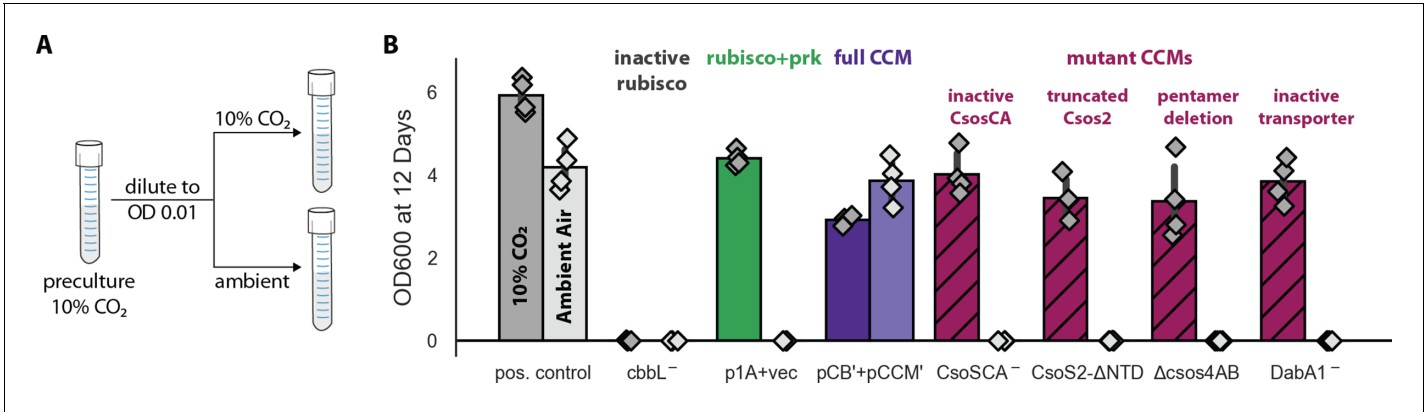

**Figure 4.** Growth in ambient air depends on the known components of the bacterial CCM. We generated plasmid variants carrying inactivating mutations to known components of the CCM. (**A**) Pre-cultures were grown in 10% $CO_2$ and diluted into pairs of tubes, one of which was cultured in 10% $CO_2$ and the other in ambient air (Materials and methods). Strains were tested in biological quadruplicate and culture density was measured after 12 days to ensure an endpoint measurement of capacity to grow. (**B**) Targeted mutations to CCM components ablated growth in ambient air while permitting growth in 10% $CO_2$, as expected. The left bar (darker color) gives the mean endpoint density in 10% $CO_2$ for each strain. The right bar (lighter color) gives the mean in ambient air. Error bars give a 95% confidence interval for the mean. From left to right, in pairs: a positive control for growth (a complemented carbonic anhydrase knockout in grey, see Materials and methods) grew in 10% $CO_2$ and ambient air, while a negative control CCMB1 strain carrying catalytically inactive rubisco (CCMB1:pCB' CbbL⁻+pCCM') failed to grow in either condition; CCMB1 expressing rubisco and prk but no CCM genes (green, CCMB1:p1A+vec) grew only in 10% $CO_2$; CCMB1:pCB'+pCCM' grew in 10% $CO_2$ and ambient air, recapitulating results presented in *Figure 3*. The following four pairs of maroon bars give growth data for strains carrying targeted mutations to CCM genes: an inactivating mutation to carboxysomal carbonic anhydrase (CCMB1:pCB' CsoSCA⁻+pCCM'), deletion of the CsoS2 N-terminus responsible for recruiting rubisco to the carboxysome (CCMB1:pCB' CsoS2 ΔNTD+pCCM'), deletion of pentameric vertex proteins (CCMB1:pCB' Δcsos4AB + pCCM'), and inactivating mutations to the DAB carbon uptake system (CCMB1:pCB' DabA1⁻ + pCCM'). All four CCM mutations abrogated growth in air while permitting growth in 10% $CO_2$. The positive control is the CAfree strain expressing human carbonic anhydrase II (Materials and methods). *Figure 4—figure supplement 1* describes statistical analyses, a 4-day replicate experiment, and additional mutants testing the contribution of rubisco chaperones to the CCM. *Figure 4—figure supplement 2* gives measurements of media pH after growth in 10% $CO_2$ and ambient air. Detailed description of all plasmid and mutation abbreviations is given in *Supplementary file 1*.

The online version of this article includes the following figure supplement(s) for figure 4:

**Figure supplement 1.** Targeted mutations to the CCM eliminate growth in ambient air.
**Figure supplement 2.** Tandem endpoint measurements of growth and culture pH.

---

*H. neapolitanus* (*Desmarais et al., 2019*), neither acRAF nor CbbQ were strictly required for CCMB1 to grow in ambient air. Consistent with our screen in the native host (*Desmarais et al., 2019*); however, *acRAF* deletion produced a substantial growth defect (*Figure 4—figure supplement 1*, panel C), suggesting that the rate of rubisco complex assembly is an important determinant of carboxysome biogenesis.

## Discussion

Today, CCMs catalyze about half of global photosynthesis (*Raven et al., 2017*), but this was not always so. Land plant CCMs, for example, arose only in the last 100 million years (*Flamholz and Shih, 2020*; *Raven et al., 2017*; *Sage et al., 2012*). Although all contemporary Cyanobacteria have CCM genes, these CCMs are found in two convergently evolved varieties (*Flamholz and Shih, 2020*; *Kerfeld and Melnicki, 2016*; *Rae et al., 2013*), suggesting that the ancestor of present-day Cyanobacteria and chloroplasts did not have a CCM (*Rae et al., 2013*). So how did carboxysome CCMs come to dominate the cyanobacterial phylum?

Here, we demonstrated that the α-carboxysome CCM from *H. neapolitanus* can be readily transferred between species and confers a large growth benefit, suggesting that these CCMs became so widespread by horizontal transfer between bacteria (*Kerfeld and Melnicki, 2016*; *Rae et al., 2013*). We constructed a functional bacterial CCM by expressing 20 genes in an *E. coli* strain, CCMB1, engineered to depend on rubisco carboxylation. In accordance with its role in native autotrophic hosts (*Desmarais et al., 2019*; *Long et al., 2018*; *Marcus et al., 1986*; *Price and Badger, 1989a*), the

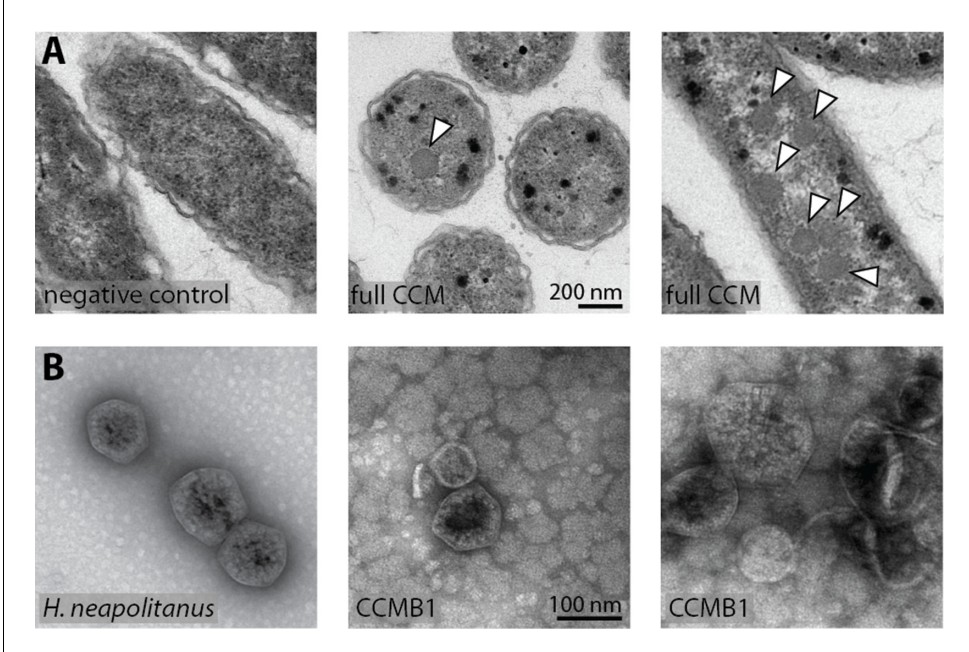

**Figure 5.** CCMB1:pCB'+pCCM' produces carboxysomes when grown in air. (**A**) Polyhedral bodies resembling carboxysomes are evident in electron micrographs of CCMB1:pCB'+pCCM' cells grown in air (full CCM, both images on the right) but were not observed in a negative control lacking pCB and pCCM plasmids (left, Methods). All panels have equal scale. (**B**) Carboxysome structures purified from CCMB1:pCB'+pCCM' grown in ambient air (Materials and methods, right) resemble structures isolated from the native host (left) in size and morphology. *Figure 5—figure supplement 2* gives full size and additional images clearly showing rubisco inside isolated carboxysomes. SDS-PAGE gels in *Figure 5—figure supplement 1* demonstrate co-migration of rubisco large and small subunits with carboxysomes structures through the purification procedure.

The online version of this article includes the following figure supplement(s) for figure 5:

**Figure supplement 1.** Carboxysomes purified from CCMB1:pCB' + pCCM' contain rubisco and other known carboxysome components.

**Figure supplement 2.** CCMB1:pCB' + pCCM' produces polyhedral bodies resembling carboxysomes when grown in ambient air.

transplanted CCM required (i) α-carboxysome structures containing both rubisco and carbonic anhydrase and (ii) inorganic carbon uptake at the cell membrane in order to enable CCMB1 to grow by fixing $CO_2$ from ambient air (*Figures 3–6*). These results conclusively demonstrate that at most 20 gene products are required to produce a bacterial CCM. The α-carboxysome CCM is apparently genetically compact and 'portable' between organisms. It is possible, therefore, that expressing bacterial CCMs in non-native autotrophic hosts will improve $CO_2$ assimilation and growth. This is a promising approach to improving plant growth characteristics (*Ermakova et al., 2020*; *Long et al., 2016*; *Wu et al., 2019*) and also engineering enhanced microbial production of fuel, food products, and commodity chemicals from $CO_2$ (*Claassens et al., 2016*; *Gleizer et al., 2019*).

Reconstitution also enabled us to test, via simple genetic experiments, whether particular genes play a role in the CCM (*Figure 4—figure supplement 1*). These experiments demonstrated that the rubisco chaperones are strictly dispensable for producing a functional bacterial CCM, although removing *acRAF* produced a substantial growth defect that warrants further investigation. Further such experiments can use our reconstituted CCM to delineate a minimal reconstitution of the bacterial CCM suitable for plant expression (*Du et al., 2014*, *Long et al., 2018*, *Long et al., 2016*; *Occhialini et al., 2016*; *Orr et al., 2020*), test hypotheses about carboxysome biogenesis (*Bonacci et al., 2012*; *Oltrogge et al., 2020*), and probe the relationship between CCMs and host physiology (*Mangan et al., 2016*; *McGrath and Long, 2014*; *Price and Badger, 1989b*). This last point deserves special emphasis as the growth physiologies of plants and bacteria are exceedingly

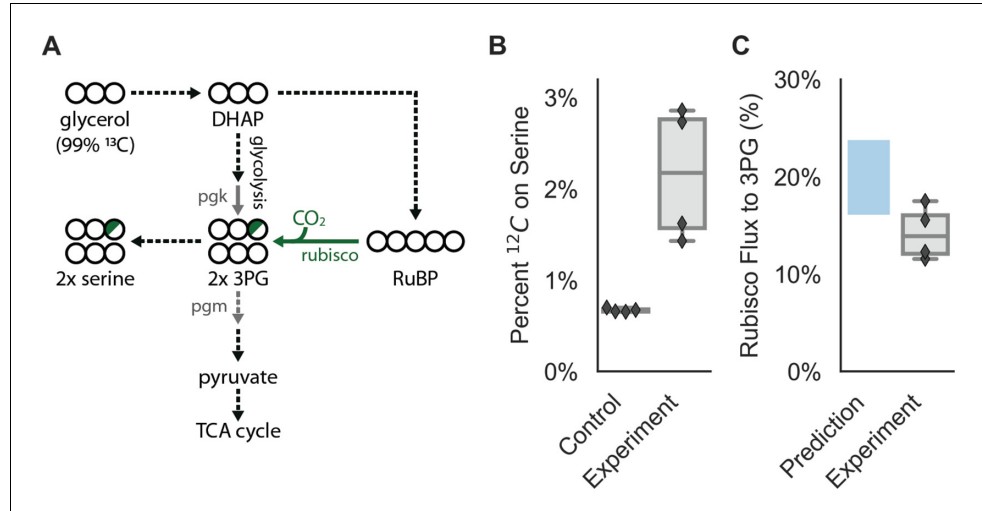

**Figure 6.** CCMB1:pCB'+pCCM' fixes $CO_2$ from ambient air into biomass. Biological replicate cultures were grown in ambient air in M9 media containing 99% $^{13}$C labeled glycerol such that $^{12}CO_2$ from air is the dominant source of $^{12}$C. In (**A**) $^{13}$C is depicted as open circles and partial $^{12}$C incorporation is indicated in green. As serine is a direct metabolic product of 3PG, we expect $^{12}$C enrichment on serine when rubisco is active in CCMB1 cells. 3PG also derives from glycolytic metabolism of glycerol, so complete $^{12}$C labeling of serine was not expected. (**B**) The $^{12}$C composition of serine from CCMB1:pCB' + pCCM' ('Experiment') is roughly threefold above the control strain (CAfree:vec+pFA-HCAII), which grows in a rubisco-independent manner (Materials and methods). *Figure 6—figure supplement 1* gives $^{12}$C composition of all measured amino acids. (**C**) The fraction of 3PG production due to rubisco was predicted via Flux Balance Analysis and estimated from isotopic labeling data (Materials and methods, Appendix 3). Estimates of the rubisco flux fraction exceeded 10% for all four biological replicates and the mean estimate of ≈ 14% accords reasonably with predictions ranging from 16 to 24%. Appendix 3 and *Figure 6—figure supplements 2–3* detail the flux inference procedure and give additional evidence for *in vivo* carboxylation from the fragmentation of serine.

The online version of this article includes the following figure supplement(s) for figure 6:

**Figure supplement 1.** Isotopic composition of amino acids from total biomass hydrolysate.

**Figure supplement 2.** $^{12}$C enrichment on serine is consistent with intracellular $CO_2$ fixation.

**Figure supplement 3.** Fragmentation of serine M+2 isotopologues confirms rubisco-catalyzed $CO_2$ addition in growing cells.

---

different and it remains unclear whether microbial CCMs can function efficiently when expressed in macroscopic land plants (*Flamholz and Shih, 2020*).

Our approach to studying CCMs by reconstitution in tractable non-native hosts can be also applied to other CCMs, including β-carboxysome CCMs, the algal pyrenoid, and plausible evolutionary ancestors thereof. Historical trends in atmospheric $CO_2$ likely promoted the evolution of CCMs (*Fischer et al., 2016*; *Flamholz and Shih, 2020*), so testing the growth of plausible ancestors of bacterial CCMs (e.g. carboxysomes lacking carbonic anhydrase activity) may provide insight into paths of CCM evolution and the composition of the ancient atmosphere at the time bacterial CCMs arose. In response to these same pressures, diverse eukaryotic algae evolved CCMs relying on micron-sized rubisco aggregates called the pyrenoids (*Flamholz and Shih, 2020*; *Wang and Jonikas, 2020*). Pyrenoid CCMs are collectively responsible for perhaps 70–80% of oceanic photosynthesis (*Mackinder et al., 2016*; *Raven et al., 2017*), yet many fundamental questions remain regarding the composition and operation of algal CCMs (*Wang and Jonikas, 2020*). Functional reconstitution of a pyrenoid CCM is a worthy goal which, once achieved, will indicate enormous progress in our collective understanding of the genetics, cell biology, biochemistry, and physical processes supporting the eukaryotic complement of oceanic photosynthesis. We hope such studies will further our principled understanding of, and capacity to engineer, the cell biology supporting $CO_2$ fixation in diverse organisms.

## Materials and methods

### Growth conditions

Unless otherwise noted, cells were grown on M9 minimal media supplemented with 0.4% v/v glycerol, 0.5 ppm thiamin ($10^4$ fold dilution of 0.5% w/v stock) and a trace element mix. The trace element mix components and their final concentrations in M9 media are: 50 mg/L EDTA, 31 mM $FeCl_3$, 6.2 mM $ZnCl_2$, 0.76 mM $CuSO_4 \cdot 5H_2O$, 0.42 mM $CoCl_2 \cdot 6H_2O$, 1.62 mM $H_3BO_3$, 81 nM $MnCl_2 \cdot 4H_2O$. 100 nM anhydrotetracycline (aTc) was used in induced cultures. For routine cloning, 25 mg/L chloramphenicol and 60 mg/L kanamycin selection were used as appropriate. Antibiotics were reduced to half concentration (12.5 and 30 mg/L, respectively) for CCMB1 growth experiments and kanamycin was omitted when evaluating rubisco-dependence of growth as pF-derived plasmids carrying kanamycin resistance also express rubisco. Culture densities were measured at 600 nm in a table top spectrophotometer (Genesys 20, Thermo Scientific) and turbid cultures were measured in five- or tenfold dilution as appropriate in order to reach the linear regime of the spectrophotometer.

Agar plates were incubated at 37°C in defined $CO_2$ pressures in a $CO_2$ controlled incubator (S41i, New Brunswick). For experiments in which a frozen bacterial stock was used to inoculate the culture, cells were first streaked on agar plates and incubated at 10% $CO_2$ to facilitate fast growth. Pre-cultures derived from colonies were grown in 2–5 mL liquid M9 glycerol media under 10% $CO_2$ with a matching 1 mL control in ambient air. Negative control strains unable to grow in minimal media (i.e. active site mutants of rubisco) were streaked on and pre-cultured in LB media under 10% $CO_2$.

Growth curves were obtained using two complementary methods: an eight-chamber bioreactor for large-volume cultivation (MC1000, PSI), and 96-well plates in a gas controlled plate reader plate (Spark, Tecan). For the 96-well format, cells were pre-cultured in the appropriate permissive media, M9 glycerol under 10% $CO_2$ where possible. If rich media was used, for example for negative controls, stationary phase cells were washed in 2x the culture volume and resuspended in 1x culture volume of M9 media with no carbon source. Cultures were diluted to an OD of 1.0 (600 nm) and 250 µl cultures were inoculated by adding 5 µl of cells to 245 µl media. A humidity cassette (Tecan) was refilled daily with distilled water to mitigate evaporation during multi-day cultivation at 37 °C. Evaporation nonetheless produced irregular growth curves (e.g. *Figure 3—figure supplement 2*), which motivated larger volume cultivation in the bioreactor, which mixes by bubbling ambient air into each growth vessel. 80 mL bioreactor cultures were inoculated to a starting OD of 0.005 (600 nm) and grown at 37°C to saturation. Optical density was monitored continuously at 680 nm.

Anaerobic cultivation of agar plates was accomplished using a BBL GasPak 150 jar (BD) flushed six times with an anoxic mix of 10% $CO_2$ and 90% $N_2$. Tenfold titers of biological duplicate cultures were plated on M9 glycerol media with and without 20 mM $NaNO_3$ supplementation. Because *E. coli* cannot ferment glycerol, $NO_3^-$ was supplied as an alternative electron acceptor. Plates without $NO_3^-$ showed no growth (*Figure 2—figure supplement 1*), confirming the presence of an anaerobic atmosphere in the GasPak.

### Computational design of rubisco-dependent strains

To computationally design mutant strains in which growth is coupled to rubisco carboxylation flux, we used a variant of Flux Balance Analysis (*Lewis et al., 2012*) called 'OptSlope' (*Antonovsky et al., 2016*). Starting from a published model of *E. coli* central metabolism, the Core *Escherichia coli* Metabolic Model (*Orth et al., 2010*), we considered all pairs of central metabolic knockouts and ignored those that permit growth in silico in the absence of rubisco and phosphoribulokinase (Prk) activities. For the remaining knockouts, we evaluated the degree of coupling between rubisco flux and biomass production during growth in nine carbon sources: glucose, fructose, gluconate, ribose, succinate, xylose, glycerate, acetate, and glycerol. This approach highlighted several candidate rubisco-dependent knockout strains, including $\Delta rpiAB$ $\Delta edd$, which is the basis of the CCMB1 strain. Full discussion of our algorithmic approach to strain design is given in Appendix 1 along with detailed description of the proposed mechanisms of rubisco coupling in CCMB1 and a comparison to other rubisc-dependent *E. coli* strains. OptSlope source code is available at https://gitlab.com/elad.noor/optslope (*Noor, 2019*) and calculations specific to CCMB1 can be found at https://github.com/flamholz/carboxecoli (*Flamholz and Noor, 2020*; copy archived at swh:1:rev:76596e1e8614173d8ef64aa13e93674307cfa3de).

## Genomic modifications producing the CCMB1 strain

Strains used in this study are documented in *Supplementary file 1*. To produce CCMB1, we first constructed a strain termed '$\Delta rpi$'. This strain has the genotype $\Delta rpiAB$ $\Delta edd$ and was constructed in the *E. coli* BW25113 background by repeated rounds of P1 transduction from the KEIO collection followed by pCP20 curing of the kanamycin selection marker (*Baba et al., 2006*; *Datsenko and Wanner, 2000*). Deletion of *edd* removes the Entner-Doudoroff pathway (*Peekhaus and Conway, 1998*), forcing rubisco-dependent metabolism of gluconate via the pentose phosphate pathway (*Figure 2—figure supplement 3*). CCMB1 has the genotype BW25113 $\Delta rpiAB$ $\Delta edd$ $\Delta cynT$ $\Delta can$ and was constructed from $\Delta rpiAB$ by deleting both native carbonic anhydrases using the same methods, first transducing the KEIO $\Delta cynT$ and then $\Delta can$ from EDCM636 (*Merlin et al., 2003*), which was obtained from the Yale Coli Genetic Stock Center. Transformation was performed by electroporation (ECM 630, Harvard Biosciences) and electrocompetent stocks were prepared using standard protocols. Strain genotypes were routinely verified by PCR, as described below.

## Recombinant expression of rubisco, prk, and CCM components

pFE21 and pFA31 are compatible vectors derived from pZE21 and pZA31 (*Lutz and Bujard, 1997*). These vectors use an anhydrotetracycline (aTc) inducible $P_{LtetO-1}$ promoter to regulate gene expression. pF plasmids were modified from parent vectors to constitutively express the tet repressor (TetR) under the $P_{bla}$ promoter so that expression is repressed by default (*Liang et al., 1999*). We found that an inducible system aids in cloning problematic genes like *prk* (*Wilson et al., 2018*). We refer to these vectors as pFE and pFA, respectively. The p1A plasmid (*Figure 2A*) derives from pFE and expresses two additional genes: the Form IA rubisco from *H. neapolitanus* and a *prk* gene from *Synechococcus elongatus PCC 7942*. The pCB plasmid is properly called pFE-CB, while pCCM is pFA-CCM. The two CCM plasmids are diagrammed in *Figure 3—figure supplement 1*. Cloning was performed by Gibson and Golden-Gate approaches as appropriate. Large plasmids (e.g. pCB, pCCM) were verified by Illumina resequencing (Harvard MGH DNA Core plasmid sequencing service) and maps were updated manually after reviewing results compiled by breseq resequencing software (*Deatherage and Barrick, 2014*). Plasmids used in this study are described in *Supplementary file 1* and available on Addgene at https://www.addgene.org/David_Savage/.

## Strain verification by PCR and phenotypic testing

As CCMB1 is a relatively slow-growing knockout strain, we occasionally observed contaminants in growth experiments. We used two strategies to detect contamination by faster-growing organisms (e.g. wild-type *E. coli*). As most strains grew poorly or not at all in ambient air, pre-cultures grown in 10% $CO_2$ were accompanied by a matching 1 mL negative control in ambient air. Pre-cultures showing growth in the negative control were discarded or verified by PCR genotyping in cases where air-growth was plausible.

PCR genotyping was performed using primer sets documented in *Supplementary file 1*. Three primer pairs were used to probe a control locus (*zwf*) and two target loci (*cynT* and *rpiA*). The *zwf* locus is intact in all strains. *cynT* and *rpiA* probes test for the presence of the CCMB1 strain (genotype BW25113 $\Delta rpiAB$ $\Delta edd$ $\Delta cynT$ $\Delta can$). Notably, the CAfree strain (BW25113 $\Delta cynT$ $\Delta can$) that we previously used to test the activity of DAB-type transporters (*Desmarais et al., 2019*) is a *cynT* knockout but has a wild-type *rpiA* locus, so this primer set can distinguish between wild-type, CAfree, and CCMB1. This was useful for some experiments where CAfree was used as a control (e.g. Figures S7-8). Pooled colony PCRs were performed using Q5 polymerase (NEB), annealing at 65°C and with a 50 s extension time.

## Selection for growth in novel conditions

CCMB1:pCB did not initially grow in M9 media supplemented with glycerol, which was unexpected because pCB carries rubisco and *prk* genes. We therefore performed a series of selection experiments to isolate plasmids conferring growth at elevated $CO_2$ and then in ambient air. Here we describe the methodology; the full series of experiments is described in Appendix 2 and illustrated in *Figure 3—figure supplement 1*. CCMB1 cultures carrying appropriate plasmids were first grown to saturation in rich LB media in 10% $CO_2$. Stationary phase cultures were pelleted by centrifugation for 10 min at 4000 x g, washed in 2x the culture volume, and resuspended in 1x culture volume of

M9 media with no carbon source. After resuspension, multiple dilutions were plated on selective media (e.g. M9 glycerol media) and incubated in the desired conditions (e.g. in ambient air) with a positive control in 10% $CO_2$ on appropriate media. When colonies formed in restrictive conditions, they were picked into permissive media, grown to saturation, washed and tested for re-growth in restrictive conditions by titer plating or streaking. Plasmid DNA was isolated from verified colonies and transformed into naive CCMB1 cells to test whether plasmid mutations confer improved growth (i.e. in the absence of genomic mutations).

We first selected for CCMB1:pCB growth on M9 glycerol media in 10% $CO_2$ and then in M9 gluconate media under 10% $CO_2$. The resulting plasmid, pCB-gg for 'gluconate grower,' was isolated and deep sequenced (Harvard MGH DNA Core plasmid sequencing service). Plasmid maps were manyally updated based on results from the breseq software (*Deatherage and Barrick, 2014*). Following this first round of selection, CCMB1 was co-transformed with pCB-gg and pCCM and selected for growth in ambient air. Washed stationary phase cultures of CCMB1:pCB-gg+pCCM were plated on M9 glycerol media in ambient $CO_2$. Parallel negative control selections were plated on uninduced plates (no aTc) and using CCMB1:p1A+pCCM, which lacks carboxysome genes. Plates were incubated in a humidified incubator for 20 days until colonies became visible.

Forty colonies were picked and tested for re-growth in ambient air by titer plating. Pooled plasmid DNA was extracted from verified colonies and electroporated into naive CCMB1 to test plasmid-linkage of growth. Colony #4 re-transformant #13 grew robustly was chosen due to replicable growth. Pooled plasmid DNA extracted from this strain was resequenced by a combination deep sequencing and targeted Sanger sequencing of the TetR locus and origins of replication, as these regions share sequence between pCB and pCCM. The individual post-selection plasmids, termed pCB' and pCCM', were reconstructed from pooled plasmid extract by PCR and Gibson cloning. These plasmids, termed pCB' and pCCM', were again verified by resequencing. Naive CCMB1 was transformed with the reconstructed post-selection plasmids pCB' and pCCM' and tested for growth in ambient air in plate reader (Spark, Tecan) and bioreactor (MC1000, PSI) assay formats.

## Design of mutant CCM plasmids

To verify that growth in ambient air depends on CCM components, we generated variants of pCB' and pCCM' carrying targeted null mutations. CCMB1 was then co-transformed with two plasmids: a mutant plasmid (of either pCB' or pCCM') and its cognate, unmodified plasmid. Mutant plasmids are listed here along with expected growth phenotypes, with full detail in *Supplementary file 1*. pCB' CbbL K194M, or pCB' cbbL⁻, contains an inactivating mutation to the large subunit of the carboxysomal Form 1A rubisco (*Andersson et al., 1989*; *Cleland et al., 1998*). This mutation was expected to abrogate rubisco-dependent growth entirely.

Mutations targeting the CCM, rather than rubisco itself, are expected to ablate growth in ambient air but permit growth in high $CO_2$. The following plasmid mutations were designed to specifically target essential components of the CCM. pCB' CsoSCA C173S, or pCB' CsoSCA⁻, carries a mutation to an active site cysteine residue responsible for coordinating the catalytic $Zn^{2+}$ ion in β-carbonic anhydrases (*Sawaya et al., 2006*). pCB' CsoS2 ΔNTD lacks the N-terminal domain of CsoS2, which is responsible for recruiting rubisco to the carboxysome during the biogenesis of the organelle (*Oltrogge et al., 2020*). Similarly, pCB' CbbL Y72R carries an arginine residue instead of the tyrosine responsible for mediating cation-π interactions between the rubisco large subunit and the N-termus of CsoS2. This mutation eliminates binding between the rubisco complex and the N-termus of CsoS2 (*Oltrogge et al., 2020*). pCB' ΔcsoS4AB lacks both pentameric shell proteins, CsoS4AB, which was shown to disrupt the permeability barrier at the carboxysome shell (*Cai et al., 2009*). pCCM' DabA1 C462A, D464A, or pCCM' DabA1⁻, carries inactivating mutations to the putative active site of the inorganic carbon transporter component DabA1 (*Desmarais et al., 2019*).

Two more mutant plasmids were designed to test the roles of rubisco chaperones in producing a functional CCM. pCCM' CbbQ K46A, E107Q, denoted pCCM' CbbQ⁻, carries mutations that inactivate the ATPase activity of the CbbQ subunit of the CbbOQ rubisco activase complex (*Tsai et al., 2015*). pCCM' ΔacRAF lacks the putative rubisco chaperone acRAF. acRAF is homologous to a plant rubisco folding chaperone (*Aigner et al., 2017*) and likely involved in the folding of the *H. neapolitanus* Form IA rubisco (*Wheatley et al., 2014*). Experimental evaluation of growth phenotypes for the above-described mutants is detailed below and results are given in *Figure 4—figure supplement 1*.

## Phenotyping of matched cultures in 10% $CO_2$ and ambient air

To interrogate the phenotypic effects of mutations to the CCM, we tested the growth of matched biological replicate cultures of CCM mutants (e.g. disruption of carboxysome components or transporter function) in M9 glycerol medium in 10% $CO_2$ and ambient air (Figure 4A). Individual colonies were picked into a round-bottom tube with 4 mL of M9 glycerol media with full strength antibiotic and 100 nM aTc. 1 mL of culture was then transferred to a second tube. The 3 mL pre-culture was incubated in 10% $CO_2$, while the 1 mL culture was incubated in ambient air as a negative control. Control strains unable to grow in minimal media (e.g. those expressing inactive rubisco mutants) were pre-cultured in LB media. High-$CO_2$ pre-cultures were grown to saturation, after which optical density (OD600) was measured in five-fold dilution. Experimental cultures were inoculated to a starting OD600 of 0.01 in 3 mL of M9 glycerol media with 12.5 mg/L chloramphenicol and 100 nM aTc. Each pre-culture was used to inoculate a matched pair of experimental cultures, one incubated in 10% $CO_2$ and another in ambient air (Figure 4A). The endpoint culture density was measured at 600 nm. All experiments were performed in biological quadruplicate. As a positive control we used a complemented double carbonic anhydrase knockout (CAfree:pFE-sfGFP+pFA-HCAII) as its growth in air depends on expression of the human carbonic anhydrase II (Desmarais et al., 2019).

## Carboxysome purification and imaging

Roughly 1.2 L of CCMB1:pCB'+pCCM' was grown in M9 glycerol media in ambient air in two identical bioreactors (MC1000, PSI) as described above. Sixteen distinct 80 mL cultures were grown, comprising eight technical replicates of two biological replicates deriving from distinct colonies. Cells were harvested before the onset of stationary phase, with optical densities ranging from $\approx 0.5$ to $\approx 2.0$ (600 nm, Genesys 20, Thermo Scientific) and pooled before subsequent purification steps. Wild type H. neapolitanus cells were grown in a 10 L bioreactor (Eppendorf BioFlo 115) modified to function as a chemostat. A continuous culture was grown in DSMZ-68 medium at a dilution rate of 0.03–0.05/hour. The culture was grown at 30°C, sparged with ambient air and the pH was held constant at 6.4 by addition of KOH. Chemostat effluent was collected in a 20 L glass flask and cells harvested every 2–3 day by centrifugation at 6000 x $g$ for 15 min. A cell pellet of approximately 10 L of culture was used for subsequent purification.

Cells were chemically lysed in B-PER II (Thermo Fischer) diluted to 1x with TEMB buffer (10 mM Tris pH 8.0, 10 mM MgCl2, 20 mM NaHCO3 and 1 mM EDTA) supplemented with 0.1 mg / mL lysozyme, 1 mM phenylmethylsulfonyl fluoride (PMSF) and 0.1 ul of benzonase/mL (Sigma-Aldrich). E. coli cells (CCMB1:pCB'+pCCM') were lysed for 30 min under mild shaking while H. neapolitanus cells were stirred vigorously with a magnetic stirrer for 1 hr. Lysed cells were centrifuged 12,000 x $g$ for 15 min to remove cell debris. The clarified lysate (supernatant) was centrifuged 40,000 x $g$ for 30 min to pellet carboxysomes and obtained pellets were gently resuspended in 1.5 mL TEMB buffer. Resuspended pellets were loaded on top of a 25 mL 10–50% sucrose step gradient (10, 20, 30, 40% and 50% w/v sucrose, made in TEMB buffer) and ultracentrifuged at 105,000 x $g$ for 35 min (SW 32 Ti Swinging-bucket, Beckman Coulter). Gradients were fractionated, analysed by SDS-PAGE and carboxysome containing fractions pooled. Due to the low concentration of carboxysomes in the CCMB1:pCB'+pCCM' sample, fraction numbers corresponding to H. neapolitanus gradient were pooled. Pooled fractions were ultracentrifuged 100,000 x $g$ for 90 min and pellets were gently resuspended in TEMB to obtain the final purified carboxysome sample. The co-migration of carboxysomes with rubisco confirmed by coomassie and silver stained SDS-page gels of the final sample. Purified carboxysomes were visualized by negative stain TEM. Sample was applied to glow discharged formvar/carbon coated copper grids. Grids were then washed with deionized water and stained with 2% aqueous uranyl acetate. Imaging was performed on a JEOL 1200 EX TEM (H. neapolitanus) or a Tecnai 12 TEM at 120 KV (FEI) (CCMB1:pCB'+pCCM'). Images were collected using UltraScan 1000 digital micrograph software (Gatan Inc).

## Thin sectioning and electron microscopy of whole cells

CCMB1:pCB'+pCCM' was grown in ambient air in 3 mL of M9 glycerol medium and induced with 100 nM aTc. A carboxysome-negative control, CAfree:pFE-sfGFP+pFA-HCAII, was grown in the same conditions. Sample preparation and sectioning were performed by the University of California Berkeley Electron Microscope Laboratory. Cell pellets were fixed for 30 min at room temperature in

2.5% glutaraldehyde in 0.1 M cacodylate buffer pH 7.4. Fixed cells were stabilized in 1% very low melting-point agarose and cut into small cubes. Cubed sample was then rinsed three times at room temperature for 10 min in 0.1 M sodium cacodylate buffer, pH 7.4 and then immersed in 1% osmium tetroxide with 1.6% potassium ferricyanide in 0.1 M cacodylate buffer for an hour in the dark on a rocker. Samples were later rinsed three times with a cacodylate buffer and then subjected to an ascending series of acetone for 10 min each (35%, 50%, 75%, 80%, 90%, 100%, 100%). Samples were progressively infiltrated with Epon resin (EMS, Hatfield, PA, USA) while rocking and later poly-merized at 60°C for 24 hr. 70 nm thin sections were cut using an Ultracut E (Leica) and collected on 100 mesh formvar coated copper grids. The grids were further stained for 5 min with 2% aqueous uranyl acetate and 4 min with Reynold's lead citrate. The sections were imaged using a Tecnai 12 TEM at 120 KV (FEI) and images were collected using UltraScan 1000 digital micrograph software (Gatan Inc).

## Sample preparation and LC-MS analysis

Protein-bound amino acids were analyzed in total biomass hydrolysate of cultures grown in minimal media with 99% $^{13}$C glycerol (Cambridge Isotopes) as the sole organic carbon source. Biological quadruplicate cultures of the experimental strain, CCMB1:pCB' + pCCM', and the rubisco-indepen-dent control strain, CAfree:pFE-sfGFP + pFA-HCAII, were grown in 80 mL volumes in a bioreactor bubbling ambient air into each growth vessel (MC1000, PSI). After harvesting biomass, samples were prepared and analyzed as described in *Antonovsky et al., 2016*. Briefly, the OD600 was recorded and 2 OD x mL of sample were pelleted by centrifugation for 15 min at 4000 x g. The pel-let was resuspended in 1 mL of 6 N HCl and incubated for 24 hr at 110°C. The acid was subsequently evaporated under a nitrogen stream using a custom gas manifold (*Nevins et al., 2005*), resulting in a dry hydrolysate. Dry hydrolysates were resuspended in 0.6 mL of MilliQ water, centrifuged for 5 min at 14,000 x g, and supernatant was analyzed by liquid chromatography-mass spectrometry (LC-MS).

Hydrolyzed amino acids were separated using ultra performance liquid chromatography (UPLC, Acquity, Waters) on a C-8 column (Zorbax Eclipse XBD, Agilent) at a flow rate of 0.6 mL/min, and eluted off the column using a hydrophobicity gradient. Buffers used were: (A) H2O + 0.1% formic acid and (B) acetonitrile + 0.1% formic acid with the following gradient: 100% of A (0–3 min), 100% A to 100% B (3–9 min), 100% B (9–13 min), 100% B to 100% A (13–14 min), 100% A (14–20 min). The UPLC was coupled online to a triple quadrupole mass spectrometer (TQS, Waters). Data were acquired using MassLynx v4.1 (Waters). Amino acids and metabolites used for analysis were selected according to the following criteria: amino acids that had peaks at a distinct retention time and m/z values for all isotopologues and also showed correct $^{13}$C labeling fractions in control samples that contained protein hydrolyzates of WT cells grown with known ratios of uniformly $^{13}$C-labeled (U-$^{13}$C) glucose to $^{12}$C-glucose. We further analyzed the serine M+2 isotopologue (parent ion in positive ion-ization mode with 108.1 m/z) using multiple reaction monitoring (MRM). This approach by selecting the channels of a daughter ion (fragment) with the formula $[C_2H_6NO]^+$: (A) 61.1 m/z, where the undetected fragment contains a carboxylic acid carbon which is a $^{13}$C isotope and (B) 62.1 m/z, where the undetected fragment contains a carboxylic acid carbon which is $^{12}$C (*Piraud et al., 2003*). We looked at the ratio of the peak integrals of A/B to infer the distribution of $^{13}$C/$^{12}$C for that partic-ular carboxyl carbon. Since the carboxylic acid on L-serine derives from the rubisco carboxylation product 3-phosphoglycerate, measuring the $^{13}$C/$^{12}$C distribution at this position reports directly on carboxylation by rubisco *in vivo* (*Figure 6* and supplements) and with lower background than the total mass measurement described above.

## Isotopic analysis of the composition of biomolecules

The total $^{13}$C fraction of each metabolite was determined as the weighted average of the fractions of all the isotopologues for that metabolite:

$$f_{13_C} = \frac{\sum_{i=0}^{N} f_i \times i}{N}$$

Here, N is the number of carbons in the compound (e.g. N = 3 for serine) and $f_i$ is the relative fraction of the i-th isotopologue, that is containing $i$ $^{13}$C carbon atoms. Each metabolite's total $^{12}$C

fraction was calculated as $f_{12c} = 1 - f_{13c}$. Our quantitative approach to inferring the rubisco carboxylation flux from these data is described fully in Appendix 3; source code and data are available at https://github.com/flamholz/carboxecoli.

## Acknowledgements

We dedicate this paper to the memory of Arren Bar-Even, who was a great friend and teacher and whose wit and intellect inspired us throughout this work. Thanks to Matt Davis for P1 transduction materials and advice, Hernan Garcia and Han Lim for pZ plasmids, Maggie Stoeva, Anna Engelbrektson, Anchal Mehra, Sophia Ewens and Tyler Barnum for help with anaerobic cultivation, Reena Zalpuri and Danielle Jorgens at the University of California Berkeley Electron Microscope Laboratory for advice and assistance with electron microscopy, and Rob Egbert and Adam Arkin for KEIO strains. We are grateful to Griffin Chure, Eric Estrin, Woody Fischer, Evan Groover, Darcy McRose, Sabeeha Merchant, Dipti Nayak, Luke Oltrogge, Naiya Phillips, and Ari Satanowski for detailed comments on the manuscript, and to Dan Arlow, Yinon Bar-On, Dan Davidi, Jack Desmarais, Hernan Garcia, Oliver Mueller-Cajar, Rob Nichols, Kris Niyogi, Dan Portnoy, Morgan Price, Noam Prywes, Jeremy Roop, Rachel Shipps, Patrick Shih, and Dan Tawfik, for support, advice and helpful discussions throughout.

## Additional information

### Competing interests

Arren Bar-Even: AB-E is co-founder of b.fab, a company aiming to commercialize engineered C1-assimilation in microorganisms. The company was not involved in this work in any way. David F Savage: DFS is a co-founder of Scribe Therapeutics and a scientific advisory board member of Scribe Therapeutics and Mammoth Biosciences. These companies were not involved in this work in any way. The other authors declare that no competing interests exist.

### Funding

| Funder | Grant reference number | Author |
|---|---|---|
| U.S. Department of Energy | DE-SC00016240 | David F Savage |
| European Research Council | NOVCARBFIX 646827 | Ron Milo |
| National Science Foundation | MCB-1818377 | David F Savage |
| Shell | EBI CW163755 | David F Savage |

The funders had no role in study design, data collection and interpretation, or the decision to submit the work for publication.

### Author contributions

Avi I Flamholz, Conceptualization, Resources, Data curation, Software, Formal analysis, Funding acquisition, Validation, Investigation, Visualization, Methodology, Writing - original draft, Project administration, Writing - review and editing; Eli Dugan, Roee Ben-Nissan, Validation, Investigation, Methodology, Writing - review and editing; Cecilia Blikstad, Data curation, Formal analysis, Investigation, Methodology, Writing - review and editing; Shmuel Gleizer, Data curation, Formal analysis, Validation, Investigation, Methodology, Writing - review and editing; Shira Amram, Resources, Investigation, Methodology; Niv Antonovsky, Conceptualization, Resources, Investigation, Methodology, Writing - review and editing; Sumedha Ravishankar, Investigation, Methodology; Elad Noor, Conceptualization, Software, Formal analysis, Investigation, Methodology, Writing - review and editing; Arren Bar-Even, Conceptualization, Resources, Supervision, Methodology, Writing - review and editing; Ron Milo, David F Savage, Conceptualization, Resources, Supervision, Funding acquisition, Methodology, Project administration, Writing - review and editing

## Author ORCIDs

Avi I Flamholz  https://orcid.org/0000-0002-9278-5479
Eli Dugan  https://orcid.org/0000-0003-2400-5511
Cecilia Blikstad  http://orcid.org/0000-0001-5740-926X
Sumedha Ravishankar  https://orcid.org/0000-0002-4026-0742
Elad Noor  https://orcid.org/0000-0001-8776-4799
Arren Bar-Even  https://orcid.org/0000-0002-1039-4328
Ron Milo  https://orcid.org/0000-0003-1641-2299
David F Savage  https://orcid.org/0000-0003-0042-2257

## Decision letter and Author response

Decision letter https://doi.org/10.7554/eLife.59882.sa1
Author response https://doi.org/10.7554/eLife.59882.sa2

# Additional files

## Supplementary files

• Supplementary file 1. This file comprises five supplementary tables. Table 1 describes the strains used in this study; Table 2 details all plasmids used; Table 3 gives primer sequences used in genotyping assays; Table 4 describes mutations observed during selection experiments; Table 5 gives a detailed description of all 20 genes expressed in this study with a detailed bibliography describing the evidence underpinning our current understanding of the molecular funciton of each gene.

• Transparent reporting form

## Data availability

All source data for all figures is available in the linked github repository along with accompanying Jupyter notebooks generating the data-driven portions of all figures.

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

## Appendix 1

## Strain design and testing

### Strain design via the OptSlope algorithm

To computationally design mutant strains in which growth is coupled to rubisco carboxylation flux, we used a variant of Flux Balance Analysis (*Lewis et al., 2012*) called 'OptSlope' (*Antonovsky et al., 2016*). Optslope searches for metabolic knockout mutants in which biomass production is coupled to flux through a reaction of choice (e.g. rubisco) at all growth rates. This coupling is evident in plots of the feasible biomass production rate against feasible rubisco fluxes. In 'rubisco-coupled' designs, maximal biomass production requires non-zero rubisco carboxylation flux and increasing biomass production demands increased carboxylation (diagrammed in *Figure 2—figure supplement 4*). The slope of this relationship is the 'coupling slope' and quantifies the degree of coupling.

Starting from a published model of *E. coli* central metabolism, the Core *Escherichia coli* Metabolic Model (*Orth et al., 2010*), we considered all pairs of central metabolic knockouts and ignored those that permit growth in silico in the absence of rubisco carboxylation (EC 4.1.1.39) and phosphoribulokinase (EC 2.7.1.19) activities. For the remaining knockouts, we evaluated the degree of coupling between rubisco flux and biomass production during growth in nine carbon sources: glucose, fructose, gluconate, ribose, succinate, xylose, glycerate, acetate, and glycerol. This approach highlighted several candidate rubisco-dependent knockout strains, including Δ*rpiAB* Δ*edd*. Consistent with the coupling mechanisms described below, OptSlope predicted rubisco-dependent growth of Δ*rpiAB* Δ*edd* strains on all carbon sources except ribose. The OptSlope algorithm is available and documented at https://gitlab.com/elad.noor/optslope, outlined in *Figure 2—figure supplement 4*, and described fully in *Antonovsky et al., 2016*. Calculations specific to CCMB1 can be found online at https://github.com/flamholz/carboxecoli.

### Proposed mechanisms of growth coupling in CCMB1

The proposed mechanism of rubisco-coupling depends on the carbon source, but in all cases coupling is explained by an inability to metabolize ribulose-5-phosphate (Ru5P) due to the removal of ribose-phosphate isomerase activity (Δ*rpiAB*). When gluconate or xylose is the growth substrate, Ru5P is produced directly from the carbon source. Although wild-type *E. coli* can metabolize gluconate via the ED pathway, the ED dehydratase knockout (Δ*edd*) in CCMB1 blocks this route and forces 1:1 production of Ru5P from gluconate. Expression of *prk* and rubisco opens a new route of Ru5P metabolism, thus enabling CCMB1 to grow in gluconate or xylose media (*Figure 2—figure supplement 3*, panel A).

When glycerol is the growth substrate, it is taken into the cell and converted to glyceraldehyde 3-phosphate, which is metabolized through lower glycolysis or gluconeogenesis. The gluconeogenesis route produces hexoses that enter the pentose phosphate pathway, which is required to synthesize ribose 5-phosphate (Ri5P) for nucleotide and histidine biosynthesis. Depending on the growth rate, products of Ri5P make up 5–25% of *E. coli* biomass (*Bremer and Dennis, 2008*; *Taymaz-Nikerel et al., 2010*). However, the pentose phosphate pathway forces co-production of Ri5P, Ru5P and xylulose 5-phosphate (Xu5P). In the absence of ribose phosphate isomerase activity, there is no pathway for metabolism of Xu5P or Ru5P. This defect is complemented by the expression of rubisco and *prk*, which together form a 'detour pathway' converting Ru5P into the lower-glycolytic metabolite 3-phosphoglycerate (*Figure 2—figure supplement 3*, panel A).

### Potential for phosphoglycolate salvage in *E. coli*

Plants, cyanobacteria and other autotrophs uniformly express 'photorespiratory' pathways to process the rubisco oxygenation product 2-phosphoglycolate, or 2PG (*Eisenhut et al., 2008*). Although these are typically called photorespiratory pathways after their discovery in plants, where they were named based on the reproducible observation of light-induced $CO_2$ production in many $C_3$ plant species (*Zelitch, 1979*), we refer to them as 'phosphoglycolate salvage pathways' following *Claassens et al., 2020* because they are also found in chemolithoautotrophic organisms lacking photosynthesis. 2PG salvage appears to be essential in plants, cyanobacteria, and

chemolithoautotrophic proteobacteria (*Claassens et al., 2020*; *Eisenhut et al., 2008*; *Somerville and Ogren, 1980*, *Somerville and Ogren, 1979*).

The *E. coli* genome encodes enzymes of a 'glycerate pathway' that could serve as a means of phosphoglycolate salvage. Indeed, this pathway was recently shown to be the primary route of phosphoglycolate salvage in *C. necator*, a proteobacterial chemolithoautotroph that notably lacks carboxysome genes (*Claassens et al., 2020*). The glycerate pathway proceeds by dephosphorylating 2PG to glycolate, oxidizing glycolate to glyoxylate, and converting two units of glyoxylate into tartronate semialdehyde via a decarboxylating lyase reaction. Tartronate semialdehyde can then be reduced to glycerate, which could enter into lower glycolysis and the TCA cycle (*Figure 2—figure supplement 3*). We attempted to delete the *gph* gene in CCMB1 as it encodes the 2PG phosphatase that catalyzes the first step of this putative pathway. However, the Δ*gph* knockout was challenging to transform by electroporation, consistent with a proposed role in DNA repair (*Teresa Pellicer et al., 2003*). We reasoned that 2PG salvage might be required in CCMB1, as photorespiratory genes are essential in cyanobacteria (*Eisenhut et al., 2008*) and chemolithoautotrophic bacteria (*Claassens et al., 2020*; *Desmarais et al., 2019*) even though both groups often express carboxysome CCMs. We therefore proceeded leaving the genes of the putative glycolate pathway intact.

## Verification of dependence on rubisco carboxylation for growth

To verify the dependence of CCMB1 on rubisco and phosphoribulokinase activities in minimal media, we constructed the p1A plasmid, expressing the large and small subunits of the *H. neapolitanus* rubisco (*cbbLS*) along with a phosphoribulokinase gene, *prk*, from the cyanobacterium *S. elongatus* PCC 7942 (*Figure 2B*). We further constructed mutant variants of p1A carrying inactive rubisco or *prk* genes. Rubisco was inactivated by mutating the large subunit active site lysine to methionine, producing p1A CbbL K194M, or p1A CbbL⁻ for short (*Andersson et al., 1989*; *Cleland et al., 1998*). Prk was inactivated by mutating ATP-binding residues in the Walker A motif, producing p1A Prk K20M S21A, termed p1A Prk⁻ for short (*Cai et al., 2014*; *Higgins et al., 1986*). CCMB1:p1A grew on glycerol and gluconate minimal media when provided 10% $CO_2$ (*Figure 2—figure supplement 5*). CCMB1:p1A CbbL⁻ and CCMB1:p1A Prk⁻ both failed to grow on minimal media supplemented with glycerol or gluconate, demonstrating a dependence on both enzymes. So long as high $CO_2$ was provided, neither activity was required for growth in rich LB media, which contains abundant nucleic acids precursors (*Sezonov et al., 2007*). We further tested five distinct bacterial rubiscos and found that they all permit robust growth in M9 glycerol media with elevated $CO_2$ (5%, *Figure 2—figure supplement 1*). Although we included the Form IC rubisco from *C. necator* (also known as *R. eutropha*), the most $CO_2$-specific bacterial rubisco known to date (*Lee et al., 1991*), none of the rubiscos tested permitted CCMB1 to grow in ambient air.

The observed high-$CO_2$ requirement of CCMB1:p1A growth was expected for two independent reasons: (i) all known rubiscos display low net carboxylation rates in ambient air due to relatively low $CO_2$ ($\approx 0.04\%$) and relatively high $O_2$ ($\approx 21\%$), as shown in *Figure 1B* and discussed in *Flamholz et al., 2019*; *Iñiguez et al., 2020*, and (ii) CCMB1 entirely lacks carbonic anhydrase activity as the open reading frames of both *cynT* and *can* genes were purposely disrupted (Materials and methods). Carbonic anhydrase knockouts of many microbes, including *E. coli* and *S. cerevisiae*, are high-$CO_2$ requiring, likely due to cellular demand for $HCO_3^-$(*Aguilera et al., 2005*; *Desmarais et al., 2019*; *Du et al., 2014*; *Merlin et al., 2003*). As the CCM cluster includes a carbonic anhydrase gene (the carboxysomal carbonic anhydrase, *csoSCA*) and the CCM is expected to qualitatively improve the carboxylation rate and $CO_2$-specificity of the encapsulated rubisco, we hypothesized that a functional CCM would enable CCMB1 to grow in ambient air.

To verify that CCMB1 depends specifically on rubisco carboxylation and not oxygenation for growth, we grew CCMB1:p1A on glycerol minimal medium in anoxic high-$CO_2$ conditions (10:90 $CO_2$:$N_2$, *Figure 2—figure supplement 2*, Materials and methods). *E. coli* predominantly respires glycerol and, therefore, grows extremely slowly on glycerol in anaerobic and low $O_2$ conditions (*Stolper et al., 2010*). We therefore supplied 20 mM $NO_3^-$ as an alternate terminal electron acceptor (*Unden and Dünnwald, 2008*) in anaerobic growth conditions (see 'Growth conditions' in Materials and methods). CCMB1:p1A grew on glycerol media in anaerobic conditions when $NO_3^-$ was provided. Growth is qualitatively weaker than a wild-type control, but this is consistent with the growth differences observed in aerobic conditions supplemented with 10% $CO_2$ (*Figure 2—figure*

supplement 2). To make this point clear, we note that *E. coli* generally grows much more robustly in the presence of $O_2$ than in its absence, as $O_2$ is the preferred terminal electron acceptor (*Unden and Dünnwald, 2008*). This can be seen by comparing the growth of the wildtype control between aerobic (10% $CO_2$, balance air) and anoxic conditions in *Figure 2—figure supplement 2*. We therefore suggest that the weak growth of CCMB1:p1A in anoxic conditions is best explained as a combination of two effects: (i) impaired growth due to removal of *rpiAB* genes, and (ii) impaired growth due to the absence of $O_2$. Alternatively, it is possible that oxygenation by rubisco plays some positive role in supporting growth, although we find this unlikely as CCMB1:p1A fails to grow in ambient air which contains abundant $O_2$, as documented in *Figure 2* and supplements. Irrespective of this nuanced issue, anaerobic growth of CCMB1:p1A on glycerol minimal media implies that growth can be supported by rubisco carboxylation alone and does not require the rubisco-catalyzed oxygenation of RuBP.

## Comparison to other rubisco-dependent *E. coli* strains

One straightforward approach to generating a rubisco selection system is to knock out rubisco in a facultative autotroph, that is a strain that can be grown in a rubisco-independent fashion for the purposes of performing genetic manipulations. This approach has been applied in the facultative chemolithoautotrophs *R. capsulatus* and *R. eutropha* and is reviewed in *Mueller-Cajar and Whitney, 2008*; *Wilson and Whitney, 2017*. Here, we focus on approaches using genetically modified *E. coli* strains to select for rubisco activity.

In addition to the CCMB1 several other strategies for coupling the growth of *E. coli* to rubisco activity have been tested. One such approach involves the deletion of glyceraldehyde 3-phosphate dehydrogenase (*gapA* gene), a core component of lower glycolysis (*Morell et al., 1992*; *Mueller-Cajar et al., 2007*). This lesion is proposed to block the production of lower glycolytic metabolites, a defect that is rescued by the expression of *prk* and rubisco which together convert pentose-phosphate pathway intermediates (ribulose 5-phosphate) into the lower glycolytic intermediate, 3-phosphoglycerate. This strain has been used to select mutant variants of the model Form II rubisco from *R. rubrum* (*Mueller-Cajar et al., 2007*). Analysis via the OptSlope algorithm predicts that the growth rate of the $\Delta gapA$ strain does not depend on the rate of carboxylation by rubisco (*Figure 1—figure supplement 1*, panel C), which might explain why all rubisco variants isolated had lower maximum carboxylation rates ($k_{cat,C}$) or lower $CO_2$ specificities ($S_{C/O}$) than the wild-type sequence.

Another approach involves co-expression of rubisco and *prk* in wildtype *E. coli*. Expression of *prk* alone entails non-productive consumption of pentose-phosphate intermediates (converting ribulose 5-phosphate into ribulose 1, 5-bisphosphate) and greatly restricts growth (*Parikh et al., 2006*; *Wilson and Whitney, 2017*). Co-expression of rubisco alleviates the negative effects of *prk* by converting the 'useless' ribulose 1, 5-bisphosphate, which is not natively found in *E. coli*, into the lower glycolytic intermediate, 3-phosphoglycerate. This approach to constructing a rubisco dependent *E. coli* strain suffers from a crucial drawback - disruption of *prk* produces a strain that grows in a rubisco-independent manner. Indeed, transposon insertions in *prk* were commonly observed and do not require rubisco for growth (*Parikh et al., 2006*; *Wilson et al., 2018*). This problem is alleviated in an improved strain, RDE2, which makes use of a *prk-neoR* fusion gene designed such that most mutations disrupting *prk* also disrupt the downstream antibiotic resistance marker. This approach greatly reduced the incidence of *prk* silencing, but did not remove it entirely (*Wilson et al., 2018*), suggesting that this approach is impractical for long-term selection experiments. Furthermore, many of the rubisco mutants produced in these strains did not display improved kinetics, but rather higher expression in *E. coli* (*Zhou and Whitney, 2019*). Expressing *prk* under a strong promoter was subsequently shown to be more deleterious to growth, which enabled selection for modest improvements in the kinetics of a bacterial rubisco (*Zhou and Whitney, 2019*). The resulting strain, termed RDE3, nonetheless produced a non-negligible number of false positives.

Considering the weaknesses of the above approaches illustrates that it is desirable (i) for the growth rate to be coupled to the rate of carboxylation by rubisco, and (ii) for this coupling to be based on *E. coli*'s native metabolism such that escape is implausible. This realization motivated the OptSlope algorithm, which we have successfully applied to design rubisco-dependent *E. coli* strains for different purposes. We previously used this approach to design strains appropriate for long-term selection experiments that successfully isolated partially and fully autotrophic *E. coli* strains, that is

strains deriving most or all of biomass carbon from an inorganic source (*Antonovsky et al., 2016*; *Gleizer et al., 2019*). The strain reported here, CCMB1, is not autotrophic. Rather, as described above and in *Figure 2—figure supplement 3*, CCMB1 relies on rubisco and Prk activities to provide a 'detour' pathway around a lesion we introduced into its metabolism by removal of ribose-phosphate isomerase activity. Since CCMB1 carries a lesion in the pentose phosphate pathway, rather than glycolysis as in our previous studies (*Antonovsky et al., 2016*; *Gleizer et al., 2019*), it is more convenient for routine work as it produces overnight colonies when grown on rich media in high $CO_2$. Since our purpose was to select for a functional CCM, and not for full autotrophy, we chose to work with CCMB1 for simplicity.

One deficiency of CCMB1, however, is that it relies on rubisco to consume ribulose 1, 5-bisphosphate and 'reconnect' its metabolism (*Figure 2—figure supplement 3*). It is therefore not required that ribulose 1, 5-bisphophate be consumed by carboxylation. Rather, rubisco-catalyzed oxygenation of ribulose 1, 5-bisphophate could, in principle, complement the $\Delta rpiAB$ lesion as the *E. coli* genome encodes enzymes that might together form a pathway metabolizing the oxygenation product, 2-phosphoglycolate (*Figure 2—figure supplement 3*, panel C). It is unlikely that this pathway is a significant contributor to the growth of CCMB1 in minimal media for several reasons that we discuss below. We also note that recently reported 'glycerate biosensor' strains might be used in future work to select specifically for rubisco carboxylation without concern for latent 2-phosphoglycolate metabolism (*Aslan et al., 2020*).

On the topic of the potential for an ersatz phosphoglycolate salvage pathway in *E. coli*, previous research suggests that the first gene of this putative pathway, 2-phosphoglycolate phosphatase, is not constitutively expressed (*Teresa Pellicer et al., 2003*). Moreover, if rubisco oxygenation was a significant contributor to the growth of complemented CCMB1 strains, we would expect growth to be linked to the presence and abundance of $O_2$. Rather, we find that these strains uniformly fail to grow in ambient air, which contains 21% $O_2$ (*Figure 2* and supplements) and grow reproducibly in anoxic media with elevated $CO_2$ (*Figure 2—figure supplement 2*) as discussed above. Finally, as shown in *Figure 2—figure supplement 1*, panel B, increasing $CO_2$ levels also increased growth rate and yield for CCMB1:p1A, implying that $CO_2$ is growth-limiting in M9 glycerol media. Altogether we conclude that CCMB1 depends on rubisco predominantly via the carboxylation of RuBP and not by its oxygenation. Nonetheless, we were careful throughout to re-transform plasmids isolated via selection experiments into naive CCMB1 in order to verify that growth phenotypes are linked to plasmid DNA. Retransformation provides some assurance that the CCM functions in the absence of any sizable genomic mutations or rearrangements that might, for example, induce expression of 2-phosphoglycolate salvage enzymes.

## Appendix 2

### Selection for growth in ambient air

#### Design of plasmids for CCM expression

As described in the Methods section, expression vectors used throughout this study were derived from pZE21 and pZA31 vectors of *Lutz and Bujard, 1997*. These two vectors have compatible origins of replication, measured copy numbers in *E. coli*, and use an anhydrotetracycline (aTc) inducible promoter to regulate gene expression. pFE and pFA plasmid backbones used herein were modified from parent vectors to constitutively express the tet repressor so that expression of heterologous gene products is repressed by default (*Liang et al., 1999*). The carboxysome expression plasmid, pCB, has a pFE backbone, while the second CCM expression plasmid, pCCM, has a pFA backbone. These genes expressed from these plasmids are diagrammed and discussed in *Figure 3—figure supplement 1*.

The expression unit of pCB derives from pHnCB10, a plasmid we previously showed enables production of carboxysome structures in *E. coli* (*Bonacci et al., 2012*). The operon expressing the carboxysome was cloned from pHnCB10 into pFE to generate pCB. Notably, this operon includes a carboxysome shell protein, csos1D, that is not natively found in the primary carboxysome operon in *H. neapolitanus* (*Bonacci et al., 2012*; *Cai et al., 2008*; *Klein et al., 2009*; *Roberts et al., 2012*). We chose to include csos1D with the carboxysome because its inclusion was previously observed to result in production of carboxysomes with more regular icosahedral morphology (*Bonacci et al., 2012*), which we hypothesized to be a correlate of proper assembly. Moreover, when we began this work, the transporters associated with proteobacterial CCMs had not yet been identified (*Desmarais et al., 2019*; *USF MCB4404L et al., 2017*; *Scott et al., 2019*) and we planned to express cyanobacterial transporters as in *Du et al., 2014*. As such, it seemed sensible to include all known carboxysome components on a single plasmid, a design we retained for convenience even after the inorganic carbon transporters were identified.

The expression unit of pCCM was cloned directly from the *H. neapolitanus* genome and encodes an operon adjacent to the carboxysome operon that expresses several CCM-related genes including a second copy of csos1D, which was left undisturbed to avoid any effects on gene expression. As diagrammed in *Figure 3—figure supplement 1* and detailed in *Supplementary file 1*, this operon encodes 10 genes including at least six with plausible roles in the CCM: a DAB type inorganic carbon transporter (dabAB1), three genes known or hypothesized to interact with rubisco (*acRAF*, *cbbOQ*) and a *parA* family gene that is likely involved in partitioning *H. neapolitanus* carboxysomes daughter cells (*MacCready et al., 2018*; *Savage et al., 2010*). For this reason, we chose to express this operon instead of the smaller DAB2 transport operon that we characterized in previous work (*Desmarais et al., 2019*).

#### Phenotypic verification of pCB and pCCM plasmids

We used reporter strains to verify the primary activities of pCB and pCCM. Previous work has shown that a carbonic anhydrase knockout strain we call CAfree is complemented by heterologous expression of carbonic anhydrases or bicarbonate transporters (*Desmarais et al., 2019*; *Du et al., 2014*; *Merlin et al., 2003*). To characterize pCCM and the DAB1 transporter it encodes, we utilized an additional plasmid, pFA-DAB1, which expresses dabAB1 and one unnamed interstitial gene (mrpA family, PFAM 00361) on their own, that is in the absence of the seven other genes natively found in the same operon. We found that both pCCM and pFA-DAB1 complement CAfree for growth in ambient air, implying that the DAB1 transport complex is functional when heterologously expressed in *E. coli* on its own or in the context of the full operon (*Figure 1—figure supplement 2*).

Given that pCB encodes a *prk* and the same rubisco as the p1A plasmid (the carboxysomal rubisco from *H. neapolitanus*, *cbbLS* genes), we expected that it would complement CCMB1 for growth on M9 glycerol media in 10% $CO_2$ as shown for CCMB1:p1A (*Figure 2*). CCMB1:pCB did not initially grow in glycerol minimal media in high $CO_2$ or ambient air, however. Since CCMB1 requires rubisco and Prk activities for growth in glycerol media (*Figure 2* and supplements) we performed a series of selection experiments to isolate plasmids conferring growth at elevated $CO_2$ and, subsequently, in ambient air. In the first round of experiments, we selected for growth of CCMB1:

pCB in minimal media in 10% $CO_2$. This produced a plasmid, termed pCB-gg, that encodes carboxysome genes and permits CCMB1 to grow in 10% CO2 on glycerol and gluconate media. We subsequently co-transformed CCMB1 with pCB-gg and pCCM to select for growth in ambient air. Plasmids isolated and reconstructed from this second round of selection experiments were termed pCB' and pCCM', which are those described in the main text and figures. Experimental protocols are described in the Materials and methods and the full series of selection experiments is diagrammed in *Figure 3—figure supplement 1*. Here, we describe selection experiments in fuller detail.

## Selection for growth of CCMB1:pCB in elevated $CO_2$

We first selected for CCMB1:pCB growth on M9 glycerol media in 10% $CO_2$ and then in M9 gluconate media under 10% $CO_2$. This was achieved by plating washed stationary phase cultures on M9 media, incubating in a humidified $CO_2$-controlled incubator, and waiting for colonies to appear. The resulting plasmid, pCB-gg.9 for 'gluconate grower #9,' was isolated and deep sequenced. pCB-gg.9 was found to carry two regulatory mutations: an amino acid substitution to the tet repressor (TetR E37A) and a nucleotide substitution in the Tet operator regulating the carboxysome operon ($tetO_2$ +8T, *Supplementary file 1*).

## Selection for growth of CCMB1:pCB gg.9+pCCM in ambient air

Following the first round of selection, CCMB1 was co-transformed with pCB-gg.9 and pCCM. Transformants grew in M9 glycerol media in 10% $CO_2$ but failed to grow on in ambient air. We therefore performed another selection experiment, plating CCMB1:pCB-gg.9+pCCM on M9 glycerol media in ambient $CO_2$. Parallel negative control selections were conducted on uninduced plates (no aTc) and using CCMB1:p1A+pCCM, which lacks carboxysome genes. Colonies formed on induced CCMB1:pCB-gg.9+pCCM plates after 20 days, but not on control plates lacking induction or carboxysome genes, respectively (*Figure 3—figure supplement 1* panel F).

　　Forty colonies were picked and tested for re-growth in ambient $CO_2$ by tenfold titer plating. 10 of 40 regrew. Six examples are given in *Figure 3—figure supplement 1* panel G. Pooled plasmid DNA was extracted from verified colonies and electroporated into naive CCMB1 to test plasmid-linkage of growth. Plasmid DNA from colony #4 produced the most robust growth in ambient air (*Figure 3—figure supplement 1* panel H). The growth of re-transformants was further evaluated by picking 16 biological replicate colonies and evaluating their growth in ambient air in liquid M9 glycerol media. Re-transformant #13 of pooled plasmid DNA from colony #4 was regrew robustly in all six technical replicates.

## Isolation and reconstruction of pCB' and pCCM'

Pooled plasmid DNA from colony #4 re-transformant #13 was resequenced by a combination deep sequencing and targeted Sanger sequencing of the TetR locus and origins of replication, as these regions share sequence between both parent plasmids (pCB and pCCM). The pCB sequence isolated from this retransformant was found to carry the same mutations as pCB-gg and pCCM had acquired the high-copy ColE1 origin of replication from pCB (*Supplementary file 1*). The individual mutant plasmids were reconstructed from pooled plasmid extract by PCR and Gibson cloning. These reconstructed post-selection plasmids, termed pCB' and pCCM', were verified once again by Illumina sequencing. Naive CCMB1 was then transformed with the reconstructed post-selection plasmids and tested for growth in ambient air. Post-selection plasmids conferred reproducible growth in ambient air in multiple growth conditions (*Figure 3*), implying that genomic mutations that formed during selections were not required to produce growth in ambient air.

## Appendix 3

### Inference of rubisco flux *in vivo*

Here, we describe our approach to estimating the rubisco flux *in vivo* in CCMB1:pCB'+pCCM' cells. As a reminder, in the experiment we performed, the organic carbon source (glycerol) is 99% $^{13}$C labeled such that inorganic carbon (e.g. $^{12}CO_2$, $H^{12}CO_3^-$) from the media is the dominant source of $^{12}$C. Our approach to estimating rubisco flux takes advantage of three observations about the central metabolism of *E. coli*.

First, we note that serine is a direct metabolic product of the rubisco carboxylation product, 3-phosphoglycerate (3PG). As such, we assume that the isotopic composition of serine, which we measure (Materials and methods), is equal to that of 3PG. Second, based on the known pathways of *E. coli* central metabolism, we presume that there are only two routes to producing 3PG in CCMB1 cells - production through lower glycolytic metabolism of glycerol via phosphoglycerate kinase (*pgk* gene) and production by rubisco. Since $^{12}$C dominantly derives from an inorganic source in our experiment (*Figure 6*) we can estimate the rubisco flux by considering the $^{12}$C labeling of serine. However, though nearly all the inorganic carbon outside the cell is $^{12}$C (natural abundance is ≈ 99%), the isotopic composition of intracellular inorganic carbon will reflect a balance of import and intracellular decarboxylation of compounds deriving from the carbon source, which is 99% $^{13}$C glycerol. The final observation is that arginine is synthesized from glutamate via a carboxylation reaction (i.e. addition of a carbamoyl phosphate deriving from $HCO_3^-$). As such, we can infer the isotopic composition of intracellular inorganic carbon by examining the difference in labeling between arginine and glutamate (*Gleizer et al., 2019*). Altogether, these observations give us a framework, described in the forgoing sections, for deriving all the information necessary to estimate the rubisco carboxylation flux *in vivo*.

### Estimating the effective intracellular $^{12}CO_2$ fraction

*E. coli* cells grown in $^{13}$C glycerol will simultaneously respire glycerol, producing intracellular $^{13}CO_2$, and take up extracellular $^{12}$C in the form of $^{12}CO_2$ and $H^{12}CO_3^-$. The isotopic composition of the intracellular inorganic carbon ($C_i$) pool will therefore reflect the balance of uptake and respiration. As rubisco carboxylation draws from the intracellular $CO_2$ pool, we must estimate the isotopic composition of the $C_i$ pool to evaluate the contribution of rubisco to 3PG and serine production. We used the carbamoyl phosphate moiety as a marker for the isotopic distribution of the intracellular $C_i$ pool, as described in *Gleizer et al., 2019*. Briefly, carbamoyl phosphate synthesis is initiated by phosphorylation of bicarbonate, and the molecule is ultimately condensed with ornithine in the biosynthesis of L-arginine. A comparison of the mass isotopologue distribution of L-arginine, which contains one carbon deriving from carbamoyl phosphate, with the mass isotopologue distribution of L-glutamate, an ornithine precursor, can thus be used to estimate the fraction of $^{13}CO_2$ in the cytosol.

We estimated the effective $^{13}$C labeling of intracellular inorganic carbon ($f_{13_{CO_2},effective}$) as follows:

$$f_{13_{CO_2},effective} = \sum_{i=0}^{6} f_{arg,i} - \sum_{i=0}^{5} f_{glu,i}$$

Here, $f_{13_{CO_2},effective}$ is the relative fraction of $^{13}CO_2$ out of the total $CO_2$ pool (or, more formally, the $C_i$ pool), and $f_{arg,i}$ and $f_{glu,i}$ are the fraction of the i-th isotopologue of arginine and glutamate respectively. We assumed fast equilibration of the intracellular $C_i$ pool because all strains used in labeling experiments express a carbonic anhydrase (either carboxysomal or cytosolic). An equivalent equation can be defined for the arginine-proline comparison (*Gleizer et al., 2019*), and we took the mean of inferences from arg-glu and arg-pro comparisons as an estimate of $f_{13_{CO_2},effective}$. The intracellular fraction of $^{12}CO_2$ was then calculated from mass balance as $f_{12_{CO_2},effective} = 1 - f_{13_{CO_2},effective}$. For brevity, we refer to these fractions as $f_{12_{CO_2}}$ and $f_{13_{CO_2}}$, respectively.

### Calculation of the intracellular rubisco carboxylation flux

When CCMB1 cells are grown on 99% $^{13}$C glycerol, 3-phosphoglycerate (3PG) can be produced via two routes: (i) rubisco-catalyzed carboxylation of RuBP and (ii) glycolytic metabolism of glycerol via

dihydroxyacetone phosphate, or DHAP (*Booth, 2005*). We denote these two fluxes as $J_{rubisco}$ and $J_{pgk}$, where pgk (phosphoglycerate kinase) is the glycolytic enzyme producing 3PG (*Bar-Even et al., 2012*). Serine is a direct metabolic product of 3PG (*Szyperski, 1995*; *Stauffer, 2004*) and was therefore assumed to have the same $^{12}$C composition as 3PG. Rubisco-catalyzed carboxylation of RuBP adds one $CO_2$ to the 5-carbon substrate, producing two 3PG molecules containing a total of six carbon atoms. Therefore, 1/6 of carbon atoms on 3PG produced via rubisco carboxylation must derive from an inorganic source (*Figure 6—figure supplement 2*). Carboxylation draws $CO_2$ from the intracellular inorganic carbon pool, whose $^{12}$C composition $f_{12_{CO_2}}$ was inferred as described above.

Based on these assumptions, the $^{12}$C composition of 3PG, and therefore serine, equals a flux-weighted sum of contributions from rubisco and pgk. As such, the relative 3PG production flux that is due to rubisco, $J_{rubisco}/(J_{rubisco}+J_{pgk})$, can be inferred via the following calculation:

$$f_{ser,ctrl} = f_{3PG,ctrl} = 0 \times \frac{1}{6}\left(f_{12_{CO_2}} + 5 \times f_{RuBP,exp}\right) + 1 \times f_{DHAP,ctrl} = f_{DHAP,ctrl}$$

$$f_{ser,exp} = f_{3PG,ctrl} = \frac{J_{rubisco}}{J_{rubisco}+J_{pgk}} \times \frac{1}{6}\left(f_{12_{CO_2}} + 5 \times f_{RuBP,exp}\right) + \frac{J_{pgk}}{J_{rubisco}+J_{pgk}} \times f_{DHAP,exp}$$

where the first equation is written for the control and the second for experimental cultures where rubisco is active (CCMB1:pCB'+pCCM'). $f_{ser,ctrl}$ and $f_{ser,exp}$ denote the $^{12}$C composition of serine in the control and experiment, respectively. Identical notation is used for RuBP and DHAP. As there are only two routes of 3PG production, the above equations can be simplified to solve for the relative flux through rubisco:

$$\frac{J_{pgk}}{J_{rubisco}+J_{pgk}} \equiv 1 - \frac{J_{rubisco}}{J_{rubisco}+J_{pgk}}$$

$$\frac{J_{rubisco}}{J_{rubisco}+J_{pgk}} = \frac{f_{ser,exp} - f_{DHAP,exp}}{\frac{1}{6}\left(f_{12_{CO_2}} + 5 \times f_{RUBP,exp}\right) - f_{DHAP,exp}}$$

To calculate the rubisco flux *in vivo* we must attach values to several parameters in the above equation. $f_{12_{CO_2}}$ was inferred on a per-sample basis, with the mean values being $20\% \pm 0.7\%$ and $61\% \pm 20\%$ for the control and experiment respectively (*Figure 6—figure supplement 3*, panel C). Because glycerol is converted into 3PG and serine via DHAP in wild-type *E. coli* (*Booth, 2005*), we expect that $f_{ser,ctrl} = f_{DHAP,ctrl}$, as derived above. LC-MS measurements give $f_{ser,ctrl} = 0.7\% \pm 0.02\%$ and $f_{ser,exp} = 2.2\% \pm 0.7\%$ (*Figure 6C*). Valine is also a metabolic product of DHAP (*Szyperski, 1995*) and was found to have a similar $^{12}$C fraction $f_{val,ctrl} = 0.7\% \pm 0.05\%$ in control cells (*Figure 6—figure supplement 1*). Since glycerol is immediately converted to DHAP in *E. coli*, we further assumed that $f_{DHAP,ctrl} = f_{DHAP,exp}$.

RuBP is produced in CCMB1 when rubisco and prk are expressed. Since glycerol is the sole carbon source and there are no carboxylation reactions between DHAP and RuBP in CCMB1, we assumed $f_{RuBP,exp} = f_{DHAP,ctrl}$. This assumption is supported by LC-MS measurements of histidine in control cells (*Figure 6—figure supplement 1*). Like RuBP, histidine is synthesized from a pentose-phosphate pathway intermediates (*Szyperski, 1995*; *Winkler and Ramos-Montañez, 2009*), and measured $f_{his,ctrl} = 0.7\% \pm 0.3\%$, which is very similar to $f_{ser,ctrl} = 0.7\% \pm 0.02\%$. Using mean values to illustrate the calculation gives $\frac{J_{rubisco}}{J_{rubisco}+J_{pgk}} = \frac{2.2\% - 0.7\%}{\frac{1}{6}(61\% + 5 \times 0.7\%) - 0.7\%} = 0.15$, implying that 15% of 3PG production is due to rubisco.

$10^5$ random samples were drawn from the experimentally determined parameter ranges to estimate a 99% confidence interval on the rubisco flux fraction. As the $^{12}$C composition of inorganic carbon ($f_{12_{CO_2}}$) and serine are mechanistically linked via rubisco, these values were assumed to co-vary. Distributions were estimated on a per-sample basis by assuming 0.1% error in direct measurement of serine and 1% error in the inference of $f_{12_{CO_2}}$. These calculations gave a median flux estimate of 15.2% with 99% of values falling between 5.0% and 23.3%. The sample with the lowest inferred rubisco flux had a median estimate of 12.3% with 99% of values falling between 3.5% and 23.9%,

implying that rubisco is responsible for a nonzero fraction of 3PG production in all samples. This and above calculations can be found on our GitHub repository in the linked Jupyter notebook.

## Predicting rubisco carboxylation flux via Flux Balance Analysis

A stoichiometric model of complemented CCMB1 was generated from the Core *Escherichia coli* Metabolic Model (*Orth et al., 2010*) by adding the *prk* and *rubisco* carboxylation reactions and then deleting *rpi* and *edd* reactions. Parsimonious Flux Balance Analysis (pFBA) was applied to the resulting model to calculate intracellular metabolic fluxes that maximize the rate of biomass production. As many distinct flux distributions can yield the same (maximal) rate of biomass production, pFBA uses the minimum sum of fluxes objective to define a unique flux solution (*Holzhütter, 2004*). The COBRApy implementation of pFBA introduces an additional free parameter, the permissible fraction of the maximal biomass production rate $f_{opt}$ (*Ebrahim et al., 2013*). When $f_{opt} < 1.0$, the biomass production can be less-than-optimal if this would further decrease the sum of fluxes. Additionally, we varied lower bound on the ATP maintenance cost, as this value affects the intracellular flux distribution and is not well-constrained in modified strains.

pFBA was run with $f_{opt}$ ranging from 0.8 to 1.0 and the lower bound on ATP maintenance ranging between 0% and 25% of ATP production to account for the fact that CCMB1 has not undergone selection to maximize biomass production. For each resulting flux distribution, the fraction of 3PG production flux due to rubisco was calculated as the fraction of 3PG molecules produced via rubisco carboxylation divided by the total flux to 3PG. These calculations predict that 16–22% of 3PG production is due to rubisco. The model was rerun after removing all possibility for product secretion by deleting all carbon-containing exchange reactions other than exchange of the carbon sources, $CO_2$. This modification should give an upper bound on the fraction of 3PG production due to rubisco, as carbon cannot be shunted away from biomass production to overflow products. The 'no overflow' model predicted that 24% of 3PG production is due to rubisco independent of $f_{opt}$. The overall range of predictions from 16–24% is plotted in *Figure 6* and supplements. All calculations were performed using Python and COBRApy (*Ebrahim et al., 2013*), and source code can be found at https://github.com/flamholz/carboxecoli.

