## [Decision Letter]

**Acceptance summary:**

Photosynthetic bacteria have evolved CO_2_ concentrating mechanism (CCM) to increase carbon fixation by Rubisco, the key photosynthetic enzyme. Savage and colleagues have now succeeded in expressing all the components necessary to establish a functional CCM in the bacterium *E. coli*. This study lays the groundwork for engineering of CCMs into crop plants for increasing yields.

**Decision letter after peer review:**

Thank you for submitting your article "Functional reconstitution of a bacterial CO_2_ concentrating mechanism in *E. coli*" for consideration by *eLife*. Your article has been reviewed by three peer reviewers, one of whom is a member of our Board of Reviewing Editors, and the evaluation has been overseen by Christian Hardtke as the Senior Editor. The following individual involved in review of your submission has agreed to reveal their identity: Martin C Jonikas.

The reviewers have discussed the reviews with one another and the Reviewing Editor has drafted this decision to help you prepare a revised submission.

We would like to draw your attention to changes in our revision policy that we have made in response to COVID-19 (https://elifesciences.org/articles/57162). Specifically, we are asking editors to accept without delay manuscripts, like yours, that they judge can stand as *eLife* papers without additional data, even if they feel that they would make the manuscript stronger. Thus, most of the revisions requested below address clarity and presentation.

Summary:

The work presented is a major scientific achievement. This is the first functional reconstitution of any CO_2_ concentrating mechanism (CCM). The work has major implications for engineering of CCMs into crops for increasing yields: the authors have definitively identified a set of components that confer CCM activity in a heterologous host. As a bonus, the authors demonstrate a new way of generating a Rubisco-dependent *E. coli*.

Revisions:

1) The EM images shown in Figure 5—figure supplement 1 should be presented as a main figure, not a supplement. The negative control is too dark and difficult to compare with the other micrographs. Moreover, it is concerning that the positive control (WT:pHnCB10) failed. It should be repeated as it would allow comparison of the putative carboxysomes to a native carboxysome and would greatly improve the quality and value of this figure.

2) For the benefit of a non-expert reader, the names of the 20 proteins and corresponding genes should listed in a table, together with their function and the relevant references.

3) In Figure 3—figure supplement 1A, the authors should discuss why the gene csos1D is present in both pCB and pCCM.

4) In Figure 4B, the large variance in the OD600 after 4 days for CCMB1:pCB'+pCCM' cultures was explained as being due to genetic effects or non-genetic differences. However, in Figure 3—figure supplement 2B the measured growth kinetics did not show such big differences. Authors please explain.

5) Would be nice if the authors can demonstrate that Rubisco localizes to the putative carboxysomes by performing an experiment such as immunogold labeling. It would improve the claim that the observed polyhedral bodies are in fact carboxysomes. We leave the decision of such an experiment to the authors.

---

## [Author Response]

Revisions:1) The EM images shown in Figure 5—figure supplement 1 should be presented as a main figure, not a supplement. The negative control is too dark and difficult to compare with the other micrographs. Moreover, it is concerning that the positive control (WT:pHnCB10) failed. It should be repeated as it would allow comparison of the putative carboxysomes to a native carboxysome and would greatly improve the quality and value of this figure.

The failure to distinguish morphological carboxysomes in the ostensible positive control is consistent with our previous publication, where excessive induction produced amorphous carboxysomes (500 μm IPTG, Bonacci et al., 2012). We accidentally used too high an induction level in this experiment and repeating the experiment with the appropriate induction level would take too long due to the COVID-19 related changes to the UC Berkeley research environment. This sample is not a true positive control in that it represents heterologous expression of carboxysome genes in an *E. coli* strain that grows in a rubisco-independent manner. We reported this data in the interest of full transparency, but it is not immediately clear to us how the failure of this particular strain to produce obvious carboxysome structures in high induction should affect the reader's interpretation of the structures seen in CCMB1:pCB’+pCCM’.

To avoid confusion, we have now removed this control from the figures. To address the reviewers interest in morphological comparison to native carboxysomes, we have included TEM images of carboxysomes we purified from *H. neapolitanus* and CCMB1 using the standard sucrose gradient purification. To address the reviewers’ concern about contrast of the negative control thin-section images, we re-stained and reimaged grids of that sample. The updated figure is now given as a standalone main text Figure 5 showing both thin-section transmission electron micrographs of cells and micrographs of purified carboxysomes.

2) For the benefit of a non-expert reader, the names of the 20 proteins and corresponding genes should listed in a table, together with their function and the relevant references.

We thank the reviewers for pointing out this unfortunate omission. We now give full detail of the gene IDs, names, descriptions, genomic location, and knockout phenotypes in Supplementary file 1—table S5 along with a list of annotated references for each gene.

3) In Figure 3—figure supplement 1A, the authors should discuss why the gene csos1D is present in both pCB and pCCM.

In *H. neapolitanus*, the csos1D gene is found at the end of the second CCM operon (diagrammed in Figure 1C). When pHnCB10 was constructed for Bonacci et al., 2011, csos1D was added to the carboxysome operon so that all the protein components of the carboxysome would be encoded on a single plasmid. This was found to yield purified carboxysomes that appear more regular on transmission EM micrographs (Figure 4A-B of Bonacci et al., 2011). Since our carboxysome plasmids derive from pHnCB10, they retain csos1D. pCCM plasmids were constructed by PCR amplification of the second operon from *H. neapolitanu*s. We chose to clone the whole operon to avoid unexpected changes to gene expression, which is why csos1D is found on both plasmids. This and other considerations associated with the design of expression plasmids for the *H. neapolitanus* CCM are now explained in full in a new appendix entitled “Appendix 2: Selection for growth in ambient air.”

4) In Figure 4B, the large variance in the OD600 after 4 days for CCMB1:pCB'+pCCM' cultures was explained as being due to genetic effects or non-genetic differences. However, in Figure 3—figure supplement 2B the measured growth kinetics did not show such big differences. Authors please explain.

We agree with the reviewers that this discrepancy is confusing. We suspect that the difference is due to variation in the lag time, as can be seen in growth curves from bioreactor and plate reader growth conditions (Figure 3—figure supplement 2 panels A-B). As a reminder, pre-cultures were all grown in high CO_2_ because (i) all control strains grow in this condition and (ii) it is faster. So ambient air growth experiments involve the dilution and transfer of a culture from 10% CO_2_ to ambient air, potentially requiring time for physiological adaptation to lower CO_2_ conditions. We note that there is much less variability in the replicate 12 day experiment report in Figure 4—figure supplement 1, implying that much of the variability in the 4 day experiment is due to variation in the duration of the lag phase. Nonetheless, from the bioreactor growth condition (Figure 3—figure supplement 2A) it seems that we should expect some variability in final growth yield between biological replicates, which indeed suggests that genetic or epi-genetic differences affect replicate phenotypes. To avoid confusing the reader, we have switched the main-text Figure 4 to give the 12-day data with lower variability.

5) Would be nice if the authors can demonstrate that Rubisco localizes to the putative carboxysomes by performing an experiment such as immunogold labeling. It would improve the claim that the observed polyhedral bodies are in fact carboxysomes. We leave the decision of such an experiment to the authors.

We would like to draw the reviewers attention to the genetic experiments in Figure 4 and Figure 4—figure supplement 1 panel C. These experiments evaluate the growth phenotypes of an N-terminal truncation of csos2 and a CbbL Y72R mutant in 10% CO_2_ and ambient air. Oltrogge et al. NSMB 2019, a recent paper from our group, showed that the rubisco-Csos2 interaction is required for rubisco to be localized to the carboxysome and that this interaction is mediated by repeat sequences in the N-terminus of Csos2. That work further demonstrated that mutating the Y72 residue of CbbL disrupts the interaction with CsoS2. Therefore, the fact that the N-terminal truncation and CbbL Y72R mutant fail to grow in ambient air (while the native sequence does grow) provides strong genetic evidence that rubisco is in fact carboxysome-localized when the CCM is expressed from un-mutated pCB’ and pCCM’. We have updated the caption of Figure 4 – supplement 1 to make this clearer.

Still, we agreed with the reviewers that it would be preferable to demonstrate this important point in an orthogonal manner. We therefore purified carboxysomes from CCMB1:pCB’+pCCM’ and wild-type *H. neapolitanus*. We imaged isolated carboxysomes by transmission electron microscopy (Figure 5B) and ran SDS-PAGE gels (Figure 5—figure supplement 2). Rubisco complexes were visible inside purified carboxysomes and both the large and small subunits were found to co-migrate with carboxysomes through the purification, implying carboxysome localization of both subunits.